# Safeguarding Visual Privacy in Dataset Distillation: Robust Initialization via Augmentation

## Abstract

Dataset distillation synthesizes small datasets that enable models to achieve accuracy comparable to training on the original full data, yielding substantial training efficiency gains. In addition, distilled data have been used for privacy-preserving applications, especially to mitigate membership inference attacks (MIA), where adversaries query a model to decide whether a sample was in its training set. However, we are the first to show that state-of-the-art dataset distillation leaks visual privacy. Distilled images can be visually consistent with private originals, as measured by LPIPS, thereby leaking sensitive information. We theoretically trace this risk to the common practice of initializing distilled images with original samples. To counter this, we propose Kaleidoscopic Transformation (KT), a plug-and-play module that applies aggregated, strong yet semantics-preserving perturbations to selected original images at initialization. Extensive experiments demonstrate that KT consistently strengthens resistance to MIA and improves visual privacy, while maintaining competitive downstream accuracy. Our code will be publicly available.

## 1 Introduction

Modern machine learning models achieve state-of-the-art performance by training on massive, high-quality datasets. At the same time, the computational and storage costs of using such large-scale data can hinder rapid model development and iteration. To address this challenge, dataset distillation (Wang et al., 2018; Guo et al., 2024) has emerged as a promising technique. It aims to synthesize a compact dataset from the original training data, such that models trained on this small synthetic set can achieve performance comparable to those trained on the entire original dataset. Current SOTA methods have demonstrated impressive performance at small scales, where they can achieve nearly lossless compression in terms of final model accuracy (Cazenavette et al., 2022; Sun et al., 2024).

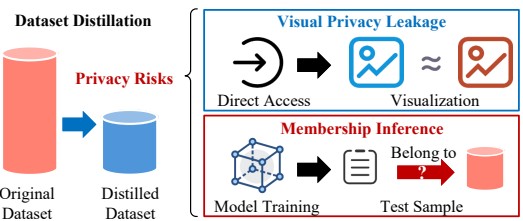

Figure 1: **The release of distilled datasets introduces new privacy risks.** Unlike model-release scenario (dark red box), where attackers employ membership inference attacks, the direct access to distilled datasets (blue box) allows for the potential visual exposure of sensitive information if the data is not adequately protected.

Beyond improving efficiency, dataset distillation benefits privacy-preserving applications by effectively defending against membership inference attacks (MIAs) (Dong et al., 2022; Carlini et al., 2022b). This capability is critical in model-release scenarios (Papernot et al., 2016), where publicly available models are vulnerable to such privacy threats. In such settings, an adversary with query access can try to determine if a specific individual's data was part of the training set (Shokri et al., 2017; Carlini et al., 2022a). By training on a distilled dataset, the resulting model has been shown to be more robust against such attacks, as it has not been directly exposed to the original private data points (Dong et al., 2022; Chen et al., 2022).

Although distilled datasets show promise in defending against MIAs, they face a new privacy threat in data-release scenarios, as illustrated in Figure 1. To mitigate membership privacy risks while still enabling efficient model training for downstream users, data curators may opt to share a distilled

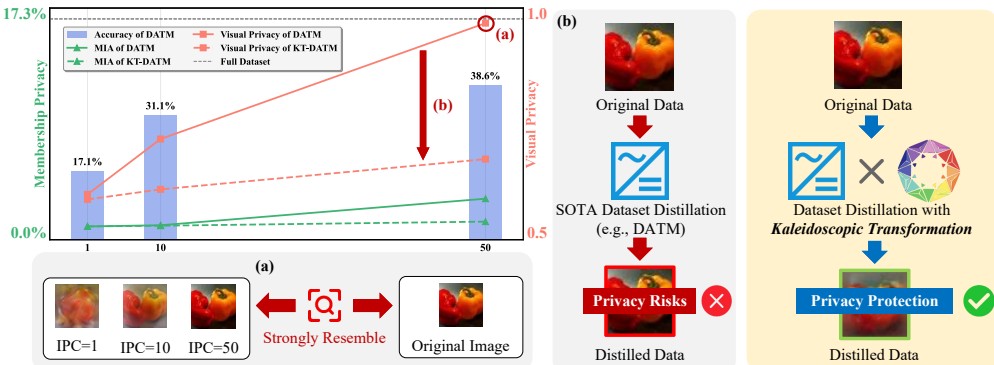

Figure 2: (a) **When IPC ∈ {1, 10, 50}, we examine the membership privacy and visual privacy leakage, comparing scenarios w/o and w/ our proposed KT.** The below show visualized distilled images corresponding to different IPC values. MIA using TPR@0.1%FPR (Carlini et al., 2022a), while visual privacy is evaluated using LPIPS (Zhang et al., 2018). (b) **Overview of the proposed KT.** As a plug-and-play module, it implements enhanced perturbations to the selected original data at the initialization phase before the distillation phase begins.

dataset instead of the original raw data. However, sharing the data creates a direct attack vector (Li et al., 2024). We term this threat **visual privacy leakage**, characterized by a strong visual similarity between distilled and original images. Attackers can visualize the images, a critical threat in sensitive domains like facial recognition (Mi et al., 2024) and medical imaging (Rui et al., 2025), where this can leak not just membership information but the sensitive content of the images themselves.

As the field of dataset distillation advances, the distilled data generated by the SOTA dataset distillation method e.g. DATM (Guo et al., 2024) *strongly resemble* to the original data, particularly with high IPC[1] (e.g., IPC = 50) , as visualized in Figure 2 (a), suggesting severely visual privacy leakage. Furthermore, visual privacy leakage can directly expose the membership of the corresponding original sample, confirming it as part of the training set and making the distilled dataset more vulnerable to MIAs at higher IPC, as depicted by the dashed green line in Figure 2 (a). Therefore, a method is needed that can maintain data utility (purple bar) at high IPC, while simultaneously preventing visual privacy leakage and enhancing the defense against MIAs.

In this study, we aim to ensure both visual privacy and membership privacy while maintaining the performance of the distilled dataset. We begin to analyze the sources of privacy leakage in dataset distillation by focusing on two phases: initialization and matching optimization. As demonstrated in Section 3.2, this leakage arises from the common practice of initializing distilled images as original data, a method known for its potential to enhance effectiveness (Yu et al., 2024; Shao et al., 2024; Cui et al., 2025). Consequently, we propose Kaleidoscopic Transformation (KT), a plug-and-play module that applies aggregated perturbations to the selected original data during the initialization phase. KT implements enhanced perturbations on these samples without engaging with the distillation process, thereby being integrated with existing SOTA dataset distillation methods, as illustrated in Figure 2 (b). KT effectively protects both membership privacy (dashed green line) and visual privacy (dashed red line) even at high IPC, while maintaining high data utility, as depicted in Figure 2 (a).

**In summary, our contribution is threefold:**

(a) To the best of our knowledge, this work is the first to explore *visual privacy leakage* in dataset distillation under high IPC settings. We further provide a theoretical explanation showing that this leakage stems from insufficient perturbation of original data initialization, which leaves the distilled samples visually aligned with their initialization counterparts.

(b) Building on this insight, we introduce a plug-and-play Kaleidoscopic Transformation (KT) module. By applying aggregated augmentations at initialization, KT protects visual information while preserving high data utility after the distillation process.

(c) Comprehensive experiments characterize the privacy leakage of existing distillation methods and validate KT's effectiveness. In specific settings, KT increases image dissimilarity by up to 25× while incurring only a 3.4% drop in utility.

---

[1] IPC means the images per class, reflecting the compression rate of the distilled dataset.

## 2 RELATED WORK

The scenario of data-release introduces new privacy risks compared to model-release scenarios. We must ensure the published data itself is privacy-protected. Attackers can directly access the data for visualization or model training. Thus, we need to integrate protection into data generation, using methods like Data Generators (Goodfellow et al., 2014) and Dataset Distillation (Wang et al., 2018).

**Privacy Preserving Data Generator.** Generative models can serve as an alternative for data sharing (Goodfellow et al., 2014). However, Chen et al. (2020) demonstrate that privacy risks exist not only when training with raw data but also when using synthetic data produced by these generative models. To address this issue, researchers have applied differential privacy (DP) (Dwork et al., 2006) to develop differentially private data generators (referred to as DP-generators) (Xie et al., 2018; Cao et al., 2021; Harder et al., 2021; Ghalebikesabi et al., 2023). However, the noise introduced by differential privacy often results in low-quality generated data, which impedes its effectiveness. Additionally, the training of DP-generators can incur significant computational costs.

**Dataset Distillation.** Dataset distillation (Wang et al., 2018) aims to improve training efficiency by extracting knowledge from a large-scale dataset and construct a significantly smaller distilled dataset, enabling models trained on it achieve comparable performance to those trained on original dataset. Current solutions can be categorized based on their optimization mechanisms (Lei & Tao, 2023), including Gradient Matching (GM) (Zhao et al., 2020; Zhao & Bilen, 2021; Kim et al., 2022), Distribution Matching (DM) (Zhao & Bilen, 2023; Yin et al., 2023; Wang et al., 2025), Trajectory Matching (TM) (Cazenavette et al., 2022; Guo et al., 2024). Remarkably, RDED (Sun et al., 2024) introduces an optimization-free paradigm, which directly crop and select realistic patches from the original data, and then stitch the selected patches into the new images as the distilled dataset. It achieves promising performance, particularly for large-scale and high-resolution datasets.

As the field progresses, state-of-the-art dataset distillation methods (Yin et al., 2023; Guo et al., 2024; Sun et al., 2024) are able to produce distilled data that achieve performance comparable to the original data. However, these distilled data closely resemble to real private data, especially at high `IPC` (e.g., `IPC` = 50). *This strong resemblance raises significant privacy concerns, necessitating urgent measures to safeguard the privacy of the distilled datasets.*

**Privacy of Distilled Dataset.** Dong et al. (2022) first build the connection between dataset distillation and differential privacy, proving that distilled data—generated via DM (Zhao & Bilen, 2023), DSA (Zhao & Bilen, 2021), and KIP (Nguyen et al., 2020)—can satisfy the definition of differential privacy. However, Carlini et al. (2022b) point out that Dong et al. (2022) incorrectly used Assumption 4.8, thus failing to provide privacy guarantees. Furthermore, recent state-of-the-art dataset distillation methods, including TM-based methods, such as MTT (Cazenavette et al., 2022), DATM (Guo et al., 2024) and non-optimization-based methods like RDED (Sun et al., 2024), have not been considered. Therefore, we focuses on examining the privacy of distilled datasets generated by these state-of-the-art distillation methods, from both theoretical and empirical perspectives in Section 3.2 and Section 4 .

## 3 PRIVACY ANALYSIS AND PROTECTION IN DATASET DISTILLATION

This section begins by introducing preliminary definitions. Subsequently, we theoretically demonstrate that the distilled dataset with high `IPC` weakens differential privacy preservation and also causes severely visual privacy leverage. Our analysis reveals that the issues predominantly arises from the common practice of initializing distilled imaegs as real data. To address these challenges, we propose a plug-and-play module, named KT, which applies expanded transformations to the selected real samples during initialization. KT ensures both differential privacy and visual privacy while maintaining the generalization performance of the distilled data.

### 3.1 PRELIMINARY

**Dataset Distillation.** Given a large-scale dataset $\mathcal{T} = \{\mathbf{x}_i, y_i\}_{i=1}^{|\mathcal{T}|}$, where $\mathbf{x}_i \in \mathbb{R}^d$ is the input sample and $y_i \in \{1, \ldots, C\}$ is the corresponding label, dataset distillation (Wang et al., 2018) aims

to synthesize a smaller distilled dataset $\mathcal{S} = \{\tilde{\mathbf{x}}_j, \tilde{y}_j\}_{j=1}^{|\mathcal{S}|}$ with $|\mathcal{S}|$ synthetic samples (i.e., $|\mathcal{S}| \ll |\mathcal{T}|$) such that models trained on $\mathcal{S}$ will have similar test performance as models trained on $\mathcal{T}$:

$$\mathbb{E}_{(\mathbf{x},y)\sim P_D} \left[ \ell \left( \phi_{\boldsymbol{\theta}_\mathcal{T}}(\mathbf{x}), y \right) \right] \simeq \mathbb{E}_{(\mathbf{x},y)\sim P_D} \left[ \ell \left( \phi_{\boldsymbol{\theta}_\mathcal{S}}(\mathbf{x}), y \right) \right] , \tag{1}$$

where $P_D$ is the test real distribution, $\mathbf{x}$ is a data sample, $\ell$ is the loss function (e.g., cross-entropy loss), and $\boldsymbol{\theta}_\mathcal{T}$ and $\boldsymbol{\theta}_\mathcal{S}$ denote the parameters of the neural network $\phi$ trained on $\mathcal{T}$ and $\mathcal{S}$, respectively.

In this paper, we decompose the dataset distillation process into two phases: initialization of the distilled data and the subsequent matching optimization, based on a review of previous studies (Guo et al., 2024). The first phase involves the initialization of distilled data, where the common strategy is to utilize real data (Yin et al., 2023; Guo et al., 2024; Sun et al., 2024). The second phase focuses on optimizing this distilled data via various matching mechanisms, as elaborated in Section 2 .

**Membership Inference Attack.**    Previous work on dataset distillation has demonstrated its defensive capabilities against MIAs in privacy-preserving applications (Dong et al., 2022; Carlini et al., 2022b). These attacks aim to determine whether a specific data point was used in training, directly impacting individual privacy (Hu et al., 2022; 2023; Niu et al., 2024).

Moreover, we conduct experiment using the state-of-the-art Likelihood Ratio Attack (LiRA) (Carlini et al., 2022a). LiRA utilizes multiple queries with various data transformations to mitigate the potential privacy-enhancing effects of data augmentation. This approach ensures a more robust evaluation of privacy risks. A detailed description of the LiRA is provided in Appendix A.2 .

**Differential Privacy (DP).**    Differential privacy (Dwork et al., 2006) introduces perturbation into the outputs to obfuscate the accurate return value, quantifying and limiting the exposure of individual information. If a mechanism can achieve DP, it can be defined as follows:

**Definition 1 (Differential Privacy) .** *A randomized mechanism $\mathcal{M}$ with range $\mathcal{R}$ is $(\epsilon, \delta)$-DP, if for any two neighboring datasets $D$ and $D'$ which differ in exactly one element, and for any subset $\mathcal{O}$ of possible outputs of $\mathcal{M}$, the following holds:*

$$\Pr[\mathcal{M}(D) \in \mathcal{O}] \leq e^\epsilon \cdot \Pr[\mathcal{M}(D') \in \mathcal{O}] + \delta . \tag{2}$$

**Visual Privacy.**    As our first contribution, we identify the risk of visual privacy leakage in dataset distillation. Unlike model-release scenarios, this risk is prominent in image data-release settings (Li et al., 2024). In this context, an attacker can directly access sensitive information by visualizing the data, a critical concern for applications involving facial or medical data. Visual privacy refers to the visual similarity between the distilled and original data, as illustrated in Figure 2 (a).

**Definition 2 (Visual Privacy) .** *For a distilled dataset $\mathcal{S}$ and a real dataset $\mathcal{T}$, visual privacy is protected if the following condition is satisfied:*

$$E(\mathcal{S}, \mathcal{T}) = \min_{\mathbf{x}_\mathcal{S} \in \mathcal{S}, \mathbf{x}_\mathcal{T} \in \mathcal{T}} \text{Dissimilarity}(\mathbf{x}_\mathcal{S}, \mathbf{x}_\mathcal{T}) > \tau , \tag{3}$$

*where $E(\mathcal{S}, \mathcal{T})$ is the worst-case dissimilarity between any two samples in $\mathcal{S}$ and $\mathcal{T}$, Dissimilarity$(\mathbf{x}_\mathcal{S}, \mathbf{x}_\mathcal{T})$ is the dissimilarity between two samples $\mathbf{x}_\mathcal{S}$ and $\mathbf{x}_\mathcal{T}$, and $\tau$ is the threshold.*

We employ the Learned Perceptual Image Patch Similarity (LPIPS) (Zhang et al., 2018) to quantify dissimilarity, thus Dissimilarity$(\mathbf{x}_\mathcal{S}, \mathbf{x}_\mathcal{T}) = \text{LPIPS}(\mathbf{x}_\mathcal{S}, \mathbf{x}_\mathcal{T})$. Unlike pixel-based metrics (Wang et al., 2004; Zhang et al., 2011), LPIPS captures the perceptual differences that are more relevant to visual privacy concerns in the distilled datasets.

**Remark 1 (Complementarity with Differential Privacy) .** *Visual Privacy complements Differential Privacy (DP) by addressing the specific risks of visual data leakage. While DP offers rigorous statistical guarantees against membership inference, it does not explicitly prevent the generation of outputs that visually resemble the original training samples (Dwork et al., 2014). Visual Privacy fills this gap by protecting data content against direct visual recognition of sensitive attributes. Therefore, a comprehensive privacy assessment for data release necessitates satisfying both membership and visual privacy metrics.*

## 3.2 PRIVACY BOUND OF MODELS TRAINED ON DISTILLED DATA

Following Dong et al. (2022), we begin by studying the privacy bound of models trained on distilled data in a differential privacy (DP) manner: *how does removing one sample in the original dataset impact models trained on distilled dataset.* It is important to highlight that our demonstration diverges from that of Dong et al. (2022) because we avoid the non-rigorous assumption in Dong et al. (2022). Our analysis focuses on the two phases of dataset distillation: the initialization of the distilled data and the matching optimization. We individually assess the differential privacy property of each phase, as elaborated in Proposition 1 and Theorem 1 .

**Phase 1: Differential privacy brought by random sampling initialization is unreliable.** To enhance the performance of distilled datasets, most dataset distillation methods use random sampling from real data as the initialization for distilled data (Sun et al., 2024; Guo et al., 2024; Yin et al., 2023). Therefore, we analyze the differential privacy guarantees of this initialization method using the following proposition.

> **Proposition 1** . *Given a training dataset of size $|\mathcal{T}|$, random sampling without replacement achieves $(\ln \frac{|\mathcal{T}|+1}{|\mathcal{T}|+1-|\mathcal{S}|}, \frac{|\mathcal{S}|}{|\mathcal{T}|})$-differential privacy, where $|\mathcal{S}|$ is the subsample size.*

This proposition suggests that random sampling initialization achieves differential privacy through randomized response (Dwork et al., 2014). (See Appendix B for proof details.)

However, an excessively large $\delta = |\mathcal{S}|/|\mathcal{T}|$ fails to satisfy the Differential Privacy (DP) requirement for $\delta$ to be on the order of $1/\|\mathcal{T}\|$. Such a $\delta$ value primarily indicates resistance to MIA rather than ensuring formal DP. It also reflects the leakage of private data used for initialization, which is proportional to the `IPC`. If subsequent distillation phases do not introduce sufficient randomness into the initialized dataset, the original initialization data could be directly exposed, motivating us to introduce the concept of visual privacy leakage.

**Phase 2: The volatility of the matching optimization introduces additional randomness to the distilled dataset, limiting individual data leakage but fully exposing initialized private data under high IPC.** The distillation process involves matching aggregated information from the original dataset, introducing randomness via iterative optimization with small batches of real data. In dataset distillation, the randomness introduced by the matching optimization is inherently applied to the initialized training samples, thereby protecting individual data information, particularly the private data used for initialization. We start by stating the objective function for matching:

$$\arg\min_{\mathcal{S}} \mathbb{E}_{\boldsymbol{\theta}_0 \sim \mathbf{P}_{\boldsymbol{\theta}}} \left[ \sum_{t=0}^{T-1} D(\xi(\mathcal{S}, \boldsymbol{\theta}^t), \xi(\mathcal{T}, \boldsymbol{\theta}^t)) \right] \tag{4}$$
$$\text{s.t.} \quad \boldsymbol{\theta}^{t+1} \leftarrow \boldsymbol{\theta}^t - \eta \cdot \nabla_{\boldsymbol{\theta}} \mathcal{L}_{\mathcal{S}}(\boldsymbol{\theta}^t).$$

Here, the function $\xi(\cdot)$ maps datasets $\mathcal{S}$ or $\mathcal{T}$ into a common space, such as gradients, features, or trajectories. The distance function $D(\cdot, \cdot)$ measures the difference between these mappings.

To analyze how this optimization process contributes to differential privacy, we focus on the distribution matching approach (Zhao & Bilen, 2023), guided by recent advancements in privacy analysis (Dong et al., 2022; Carlini et al., 2022b). In their analysis, Dong et al. (2022) reveals the relationship between the finnal distilled dataset $\mathcal{S}^*$ and the original dataset $\mathcal{T}$, as shown in the following lemma:

> **Lemma 1 (Connection between $\mathcal{S}^*$ and $\mathcal{T}$ (Dong et al., 2022)) .** *For a real data initialization, if the optimized distilled dataset $\mathcal{S}^*$ is derived from $\mathcal{S} = \mathbf{s}_1, \cdots, \mathbf{s}_{|\mathcal{S}|}$ through distribution matching, then:*
> $$\mathbf{s}_i^* = \mathbf{s}_i + \frac{1}{|\mathcal{T}|} \sum_{j=1}^{|\mathcal{T}|} \mathbf{x}_j - \frac{1}{|\mathcal{S}|} \sum_{j=1}^{|\mathcal{S}|} \mathbf{s}_j \in span(\mathcal{T}), \tag{5}$$
> *where $span(\mathcal{T}) := \{\sum_{i=1}^{|\mathcal{T}|} w_i \mathbf{x}_i | 1 \le i \le |\mathcal{T}|, w_i \in \mathbb{R}, \mathbf{x}_i \in \mathcal{T}\}$ denotes the linear span of the dataset $\mathcal{T}$.*

**Remark 2 .** *This lemma demonstrates that the distilled dataset $\mathcal{S}^*$, when derived through optimized matching, closely aligns with the distribution of $\mathcal{T}$. The proximity of $\mathcal{S}^*$ to $\mathcal{T}$ implies that as the size of $\mathcal{S}$ approaches that of $\mathcal{T}$, the distilled samples $\mathbf{s}_i^*$ resemble the original samples $\mathbf{s}_i$, thereby potentially increasing visual privacy risks, as shown in Figure 2 (a).*

*The distilled dataset, derived through optimized matching from the initial data, can be conceptualized as a normal distribution with $\mu = \mathbf{s}_i + 1/|\mathcal{T}| \sum_{j=1}^{|\mathcal{T}|} \mathbf{x}_j - 1/|\mathcal{S}| \sum_{j=1}^{|\mathcal{S}|} \mathbf{s}_j$. Consequently, by comparing the Kullback-Leibler divergence between adjacent datasets, we can ascertain the privacy protection capabilities of the distilled dataset.*

Building upon Lemma 1, we utilize the concept of adjacent datasets from DP to compare distributional differences. Our analysis reveals that the matching stage limits the influence of any single training sample on the distilled result, thereby protecting membership privacy, as formalized in the following theorem (see our proof details in Appendix C):

**Theorem 1 .** *Consider a target dataset $\mathcal{T}$ and a leave-one-out adjacent dataset $\mathcal{T}' = \mathcal{T} \setminus \{\mathbf{x}\}$, where $\mathbf{x}$ is not sampled for initialization in phase 1. The distilled datasets $\mathcal{S}$ and $\mathcal{S}'$, with $|\mathcal{S}| = |\mathcal{S}'| \ll |\mathcal{T}|$, show that the membership privacy leakage from removing $\mathbf{x}$ is bounded by:*

$$D_{\mathrm{KL}}(P \parallel Q) \leq \frac{2B|\mathcal{S}|}{|\mathcal{T}|} \cdot \lambda_{\max}(\mathbf{\Sigma}^{-1}),$$

*where $P$ and $Q$ are the sample distributions of the distilled datasets $\mathcal{S}$ and $\mathcal{S}'$, respectively, $B$ is the upper bound value of the original data and $\lambda_{\max}$ is the largest eigenvalue of the inverse covariance matrix $\mathbf{\Sigma}$.*

Theorem 1 states that the privacy leakage introduced by the matching optimization is limited. However, it is important to note that while the matching process itself offers some privacy protection, the initialization phase can still pose initial data privacy risks. Notably, the majority of state-of-the-art distillation methods (Cazenavette et al., 2022; Guo et al., 2024; Sun et al., 2024) employ initialization with original data to improve performance, which leads to a significant privacy concern.

### 3.3 METHOD FOR VISUAL PRIVACY PROTECTION

As previously discussed, although dataset distillation can theoretically limit the leakage of individual data, initializing training samples can significantly expose privacy risks, especially under high IPC conditions, leading to visual privacy leakage. To address this issue, we propose a plug-and-play module, termed Kaleidoscopic Transformation (KT), which introduce strong transformations to the selected real data during initialization. This module builds upon Differentiable Siamese Augmentation (DSA) (Zhao & Bilen, 2021), a promising approach originally designed to improve the generalization capabilities of distilled datasets. In our study, we adapt DSA as a transformation technique applied to

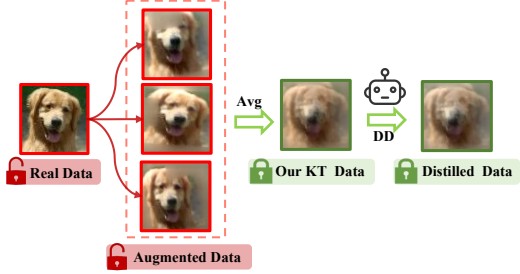

Figure 3: **Overview of Kaleidoscopic Transformation (KT).** We generate multiple augmented samples for each single input and average them to obtain the final strongly augmented sample.

the initialized real private data. The randomness introduced by these transformations enhances the differential privacy property of the distilled dataset and provides better visual privacy protection.

**Kaleidoscopic transformation.** Consider the set $\mathcal{A}$ of all differentiable augmentations. Assume we have a sequence of image transformations $\{T_1, \ldots, T_i, \ldots, T_m\} \subset \mathcal{A}$, such as rotation, with each transformation $T_i$ associated with a probability $p_i$ of being executed. By leveraging these augmentations, we can generate a newly augmented dataset. The $j$-th augmented sample of the $i$-th example is:

$$\mathbf{s}'_{i,j} = \left( \circ_{k=1}^m T_k^{U_{i,j,k} \leq p_k} \right) (\mathbf{s}_i), \tag{6}$$

For transformation $T_i$, we generate a random variable $U_i \sim \text{Uniform}(0, 1)$. If $U_i \leq p_i$, $T_i$ is applied to the input image.

To enhance the transformation process, we produce $n$ augmented samples for each input and derive the final augmented sample by averaging: $\mathbf{s}'_i = \frac{1}{n} \sum_{j=1}^{n} \mathbf{s}'_{i,j}$. As illustrated in Figure 3, employing multiple data augmentations can substantially improve privacy protection. Therefore, we initialize the distilled dataset using transformed samples $\mathbf{s}'$, rather than the original samples $\mathbf{s}$.

Note that KT not only enhances visual privacy of the distilled dataset but also introduces additional randomness into the distillation process, thereby strengthening the resistance to MIAs. We justify this by modeling a differential transformation as a random bounded perturbation $\boldsymbol{\epsilon}$ (Rajput et al., 2019), with $\|\boldsymbol{\epsilon}\| \leq \epsilon_0$ and $\|T(\mathbf{s}) - \mathbf{s}\| \leq \epsilon_0$. It allows modeling the distribution of the distilled dataset obtained through KT, therefore enabling calculating the KL divergence between adjacent datasets in Theorem 2 (see proof details in Appendix D):

> **Theorem 2.** *Consider a target dataset $\mathcal{T}$ and a leave-one-out dataset $\mathcal{T}' = \mathcal{T} \setminus \mathbf{x}$, where $\mathbf{x}$ is not used for initialization in phase 1. The KT initialized distilled datasets $\mathcal{S}_{\mathrm{KT}}$ and $\mathcal{S}'_{\mathrm{KT}}$, with $|\mathcal{S}_{\mathrm{KT}}| = |\mathcal{S}'_{\mathrm{KT}}| \ll |\mathcal{T}|$, show that the membership privacy leakage from removing $\mathbf{x}$ is bounded by:*
>
> $$\mathrm{D}_{\mathrm{KL}}(P_{\mathrm{KT}} \parallel Q_{\mathrm{KT}}) \leq \frac{2B\,|\mathcal{S}|}{|\mathcal{T}|} \cdot \lambda_{\max}((\boldsymbol{\Sigma} + {1}/{n}\boldsymbol{\Sigma}_{\boldsymbol{\epsilon}})^{-1})$$
>
> *where $P_{\mathrm{KT}}$ and $Q_{\mathrm{KT}}$ are the sample distributions of the distilled datasets $\mathcal{S}_{\mathrm{KT}}$ and $\mathcal{S}'_{\mathrm{KT}}$.*

## 4 EXPERIMENT

### 4.1 EXPERIMENT SETUP

**Datasets and Neural Networks:** We conduct experiments on both small-scale and large-scale datasets. For small-scale data, we evaluate our method on CIFAR-10 ($32 \times 32$) (Krizhevsky et al., 2009b) and CIFAR-100 ($32 \times 32$) (Krizhevsky et al., 2009a). For large-scale data, we conduct experiments on Tiny-ImageNet ($64 \times 64$) (Le & Yang, 2015), to assess the scalability and effectiveness of our method on more complex and varied datasets.

Following previous dataset distillation studies (Yin et al., 2023; Sun et al., 2024; Guo et al., 2024), we employ ConvNet (Guo et al., 2024) as our backbone architectures across all datasets. For ConvNet, Conv-3 is employed for CIFAR-10/100, while Conv-4 is used for Tiny-ImageNet.

We also conduct additional experiments on privacy-sensitive datasets, i.e., CelebA (Liu et al., 2015) and COVID-19 (Li et al., 2022), as well as cross-architecture evaluations in Appendix J.

**Baselines:** We evaluate our proposed method, KT, against a range of state-of-the-art techniques in both dataset distillation and data generator. For all experiments, we utilize three different random seeds and report both the mean and variance of the results.

- Dataset Distillation Methods: (1) distribution matching-based methods, such as DM (Zhao & Bilen, 2023); (2) gradient matching-based methods, exemplified by DSA (Zhao & Bilen, 2021); (3) trajectory matching-based strategies, including MTT (Cazenavette et al., 2022) and DATM (Guo et al., 2024); and (4) non-optimization-based frameworks like RDED (Sun et al., 2024).
- Data Generator Methods: (1) DP generator, such as DP-MEPF (Harder et al., 2022); (2) DP distillation-based methods, such as PSG (Chen et al., 2022).

**MIA Settings and Privacy Metrics:** We consider a typical scenario where the adversary possesses access to the distilled dataset $\mathcal{S}$ and employs it to train a target model $f_{\mathcal{S}}$. The objective of adversary is to infer membership information of the original dataset $\mathcal{T}$.

For our MIA framework on distilled datasets, we consider the entire original training set as members of the distilled dataset, as all samples contribute to the distillation process. To ensure fairness, we employ identical test samples and shadow models across various distilled and original datasets (see Appendix E.3 for a detailed illustration of our framework). Following Carlini et al. (2022a), we use TPR @ 0.1% FPR as the success criterion for MIAs.

Table 1: Comparison with previous dataset distillation methods on CIFAR-100 and Tiny ImageNet. **Membership Privacy** and **Visual Privacy** are evaluated via TPR@0.1% FPR and LPIPS, respectively.

| | Method | TPR@0.1%FPR (↓) | | | Min LPIPS (↑) | | | Test Accuracy (↑) | | |
|---|---|---|---|---|---|---|---|---|---|---|
| | | 1 | 10 | 50 | 1 | 10 | 50 | 1 | 10 | 50 |
| CIFAR-100 | Full Dataset | | $24.8 \pm 0.4^*$ | | | $0^*$ | | | $61.27^*$ | |
| | DM | $0.11 \pm 0.02$ | $0.18 \pm 0.01$ | $0.9 \pm 0.1$ | 0.35 | 0.26 | 0.14 | $11.4 \pm 0.3$ | $29.7 \pm 0.3$ | $43.6 \pm 0.4$ |
| | KT-DM | $0.11 \pm 0.01$ | $0.16 \pm 0.02$ | $0.42 \pm 0.05$ | 0.40 | 0.33 | 0.31 | $7.8 \pm 0.1$ | $24.1 \pm 0.2$ | $40.2 \pm 0.3$ |
| | DSA | $0.11 \pm 0.02$ | $0.19 \pm 0.01$ | $1.3 \pm 0.1$ | 0.34 | 0.24 | 0.16 | $13.9 \pm 0.4$ | $32.4 \pm 0.3$ | $38.6 \pm 0.3$ |
| | KT-DSA | $0.1 \pm 0.03$ | $0.17 \pm 0.02$ | $0.45 \pm 0.03$ | 0.39 | 0.33 | 0.32 | $8.2 \pm 0.3$ | $26.5 \pm 0.2$ | $35.3 \pm 0.2$ |
| | MTT | $0.1 \pm 0.02$ | $0.19 \pm 0.05$ | $1.8 \pm 0.1$ | 0.31 | 0.19 | 0.08 | $24.3 \pm 0.3$ | $39.7 \pm 0.4$ | $47.7 \pm 0.2$ |
| | KT-MTT | $0.1 \pm 0.02$ | $0.16 \pm 0.02$ | $0.5 \pm 0.2$ | 0.39 | 0.33 | 0.30 | $22.1 \pm 0.2$ | $34.6 \pm 0.3$ | $42.8 \pm 0.3$ |
| | DATM | $0.13 \pm 0.03$ | $0.4 \pm 0.05$ | $\mathbf{3.2 \pm 0.1}$ | 0.29 | 0.14 | $\mathbf{0.01}$ | $27.9 \pm 0.2$ | $47.2 \pm 0.4$ | $\mathbf{55.0 \pm 0.2}$ |
| | KT-DATM | $0.1 \pm 0.02$ | $0.16 \pm 0.02$ | $\mathbf{0.6 \pm 0.2}$ | 0.31 | 0.29 | $\mathbf{0.29}$ | $22.8 \pm 0.2$ | $40.2 \pm 0.3$ | $\mathbf{49.2 \pm 0.3}$ |
| | RDED | $0.14 \pm 0.02$ | $0.44 \pm 0.05$ | $\mathbf{3.4 \pm 0.1}$ | 0.04 | 0.02 | $\mathbf{0.01}$ | $19.6 \pm 0.3$ | $48.1 \pm 0.3$ | $\mathbf{57.0 \pm 0.1}$ |
| | KT-RDED | $0.1 \pm 0.02$ | $0.17 \pm 0.01$ | $\mathbf{0.6 \pm 0.06}$ | 0.28 | 0.28 | $\mathbf{0.27}$ | $13.2 \pm 0.4$ | $40.2 \pm 0.3$ | $\mathbf{54.1 \pm 0.5}$ |
| Tiny-ImageNet | Full Dataset | | $17.3 \pm 0.5^*$ | | | $0^*$ | | | $49.73^*$ | |
| | DM | $0.1 \pm 0.02$ | $0.15 \pm 0.05$ | $0.9 \pm 0.2$ | 0.34 | 0.31 | 0.21 | $3.9 \pm 0.2$ | $12.9 \pm 0.4$ | $24.1 \pm 0.3$ |
| | KT-DM | $0.1 \pm 0.02$ | $0.15 \pm 0.02$ | $0.3 \pm 0.04$ | 0.42 | 0.32 | 0.32 | $2.2 \pm 0.2$ | $9.1 \pm 0.2$ | $22.7 \pm 0.3$ |
| | DSA | – | – | – | – | – | – | – | – | – |
| | KT-DSA | – | – | – | – | – | – | – | – | – |
| | MTT | $0.1 \pm 0.02$ | $0.17 \pm 0.04$ | $1.1 \pm 0.2$ | 0.36 | 0.18 | 0.04 | $8.8 \pm 0.3$ | $23.2 \pm 0.2$ | $28.0 \pm 0.3$ |
| | KT-MTT | $0.1 \pm 0.02$ | $0.16 \pm 0.02$ | $0.5 \pm 0.2$ | 0.36 | 0.31 | 0.25 | $7.8 \pm 0.2$ | $20.4 \pm 0.1$ | $24.7 \pm 0.2$ |
| | DATM | $0.12 \pm 0.08$ | $0.2 \pm 0.04$ | $\mathbf{2.4 \pm 0.1}$ | 0.32 | 0.10 | $\mathbf{0.01}$ | $17.1 \pm 0.3$ | $31.1 \pm 0.3$ | $\mathbf{38.6 \pm 0.3}$ |
| | KT-DATM | $0.1 \pm 0.02$ | $0.16 \pm 0.02$ | $\mathbf{0.5 \pm 0.2}$ | 0.34 | 0.26 | $\mathbf{0.25}$ | $13.3 \pm 0.2$ | $27.6 \pm 0.3$ | $\mathbf{35.2 \pm 0.3}$ |
| | RDED | $0.12 \pm 0.04$ | $0.23 \pm 0.02$ | $\mathbf{2.8 \pm 0.1}$ | 0.04 | 0.02 | $\mathbf{0.01}$ | $12.0 \pm 0.1$ | $39.6 \pm 0.1$ | $\mathbf{49.6 \pm 0.2}$ |
| | KT-RDED | $0.11 \pm 0.01$ | $0.18 \pm 0.02$ | $\mathbf{0.6 \pm 0.07}$ | 0.25 | 0.23 | $\mathbf{0.20}$ | $7.6 \pm 0.3$ | $33.5 \pm 0.2$ | $\mathbf{47.3 \pm 0.2}$ |

Table 2: Comparison with previous **DP-data generation methods** on CIFAR-10.

| | Method | TPR@0.1%FPR (↓) | | | Min LPIPS Distance (↑) | | | Test Accuracy (↑) | | |
|---|---|---|---|---|---|---|---|---|---|---|
| | | 1 | 10 | 50 | 1 | 10 | 50 | 1 | 10 | 50 |
| CIFAR-10 | DP-MEPF($\epsilon = 10$) | $0.1 \pm 0.01$ | $0.13 \pm 0.01$ | $0.16 \pm 0.02$ | 0.40 | 0.38 | 0.35 | $16.6 \pm 0.4$ | $24.1 \pm 0.3$ | $28.0 \pm 0.2$ |
| | PSG($\epsilon = 10$) | $0.1 \pm 0.02$ | $0.12 \pm 0.03$ | $0.15 \pm 0.04$ | 0.42 | 0.38 | 0.34 | $28.9 \pm 0.4$ | $40.3 \pm 0.5$ | $47.2 \pm 0.2$ |
| | KT-DATM | $0.1 \pm 0.02$ | $0.14 \pm 0.02$ | $0.4 \pm 0.1$ | 0.36 | 0.35 | 0.33 | $43.3 \pm 0.2$ | $62.3 \pm 0.1$ | $\mathbf{69.2 \pm 0.2}$ |
| | KT-RDED | $0.12 \pm 0.01$ | $0.18 \pm 0.03$ | $0.7 \pm 0.1$ | 0.35 | 0.34 | 0.31 | $17.7 \pm 0.2$ | $42.2 \pm 0.2$ | $\mathbf{62.5 \pm 0.3}$ |

For visual privacy, we use Min LPIPS (Zhang et al., 2018) to evaluate the visual similarity between distilled and original data. A larger LPIPS signifies stronger visual privacy, meaning that no distilled sample is visually similar to any sample in the original dataset. Other perceptual similarity metrics such as SSCD (Pizzi et al., 2022) and DreamSim (Fu et al., 2023) can be used interchangeably. Additional results are provided in Appendix I.

Further comprehensive experimental configurations, including detailed settings aligned with the original distillation methods are provided in Appendix E.

## 4.2 DENFENSE OF DISTILLED DATASETS AGAINST MIA

**Comparison with SOTA Dataset Distillation Methods.** We use TPR@0.1% FPR (Carlini et al., 2022a) to evaluate the differential privacy of distilled datasets, focusing on attack success at low false positive rates. It is evident that LiRA successfully attacks all three full datasets, as shown in Table 1. However, models trained on distilled datasets, even without employing the our KT method, substantially reduces the attack success rate. *The results confirms that distilled datasets can denfense against MIA, aligning with our analysis in Section 3.2.* Notably, when KT is applied, the attack success rate decrease, further verifying that KT enhances the defense against MIA. Detailed results for CIFAR-10 can be found in Appendix Appendix F.

**Comparison with SOTA Data Generator Methods.** We further compare our method KT with existing data generation techniques designed for differential privacy and visual privacy, as illustrated in Table 2. Our experiments focus on CIFAR-10, as it is the primary benchmark for most DP data generation methods. Other datasets like Tiny-ImageNet are often treated as public data by some methods (Wang et al., 2024; Lin et al., 2023).

*Our approach demonstrates a balanced performance in privacy preservation and data utility.* While methods like PSG and DP-MEPF exhibit strong privacy guarantees due to their strict privacy budgets

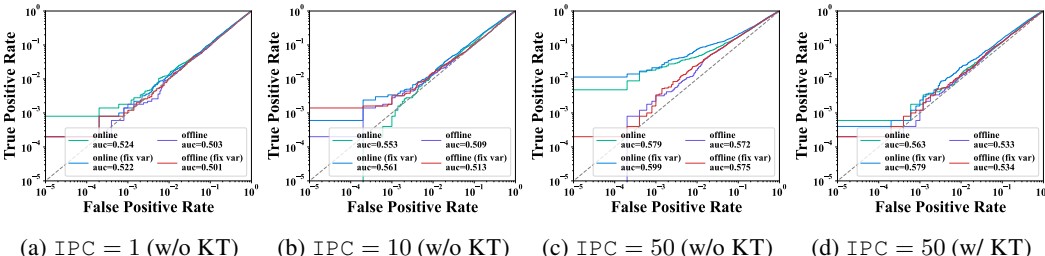

(a) IPC = 1 (w/o KT)    (b) IPC = 10 (w/o KT)    (c) IPC = 50 (w/o KT)    (d) IPC = 50 (w/ KT)

Figure 4: ROC curve graphs of DATM on TinyImageNet at different IPC values: With higher IPC, the success rate of attacks at low false positive rates increases. The application of KT at IPC = 50 demonstrates a significant reduction in attack success rate.

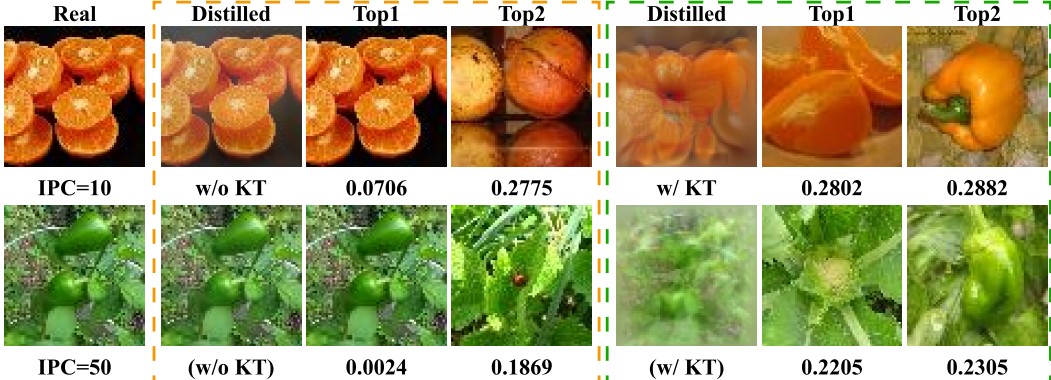

Figure 5: **DATM presents visual privacy protection at IPC=10 and 50.** The orange and green regions represent the distilled samples without and with the KT plugin, respectively. We selected the top 2 distilled data that are most similar to the original data, with the values measured using LPIPS.

and noise initialization, they struggle with data utility, particularly in downstream tasks requiring model training with low IPC in Appendix G.

**Impact of Varying IPC on Resisting MIA.** We apply LiRA with results illustrated in Figure 4 to demonstrate the impact of IPC changes on MIA resistance. As the IPC value increases, AUC of LiRA's ROC curves show also increase, which suggests that higher IPC values reduce the differential privacy protection of the distilled datasets.

Furthermore, for a high IPC of 50, we compare scenarios with and without our KT. The results presented in Figure 4 (c) and (d), show that our KT reduces the AUC scores of the ROC curves, demonstrating that *our* KT *effectively enhances differential privacy, even at elevated IPC levels.*

### 4.3 ENHANCED VISUAL PRIVACY VIA KT

**Comparison with SOTA Methods.** Beyond membership privacy, dataset distillation in data-release settings poses a visual privacy leakage risk. We therefore assess visual privacy for standard distillation methods and for the same methods augmented with KT. Additionally, we conducted experiments on the open COVID-19 CXR dataset (Li et al., 2022) which is sensitive to privacy in Appendix J.3.

The results in Table 1 indicate that a higher IPC leads to a notable decrease in the Min LPIPS. *This suggests that higher IPC more severely exposure visual privacy, consistent with our analysis in Section 3.2.*

Furthermore, we visualize samples of the distilled dataset and the top-2 nearest samples from the original dataset in Figure 5. At IPC = 10 and 50, the distilled dataset without KT completely leaks the original data, indicating significant visual privacy leakage. *With the introduction of KT, the distilled samples are visually distinct from their nearest neighbors in the original dataset, demonstrating enhanced visual privacy.*

**Influence of Hyper-parameter $n$.** The choice of $n$ allows users to navigate the privacy-utility trade-off. To determine the optimal setting for the hyper-parameter $n$, we conducted experiments varying $n$ from 1 to 5 with KT-DATM on TinyImageNet using `IPC = 50` in Figure 6. *Our findings reveal a critical trade-off between privacy protection and data utility.* Data publishers should set a target Visual Privacy threshold $\tau$ and select the smallest $n$ that satisfies this condition on a validation set. Empirically, $n = 3$ serves as a robust universal default across various datasets and algorithms.

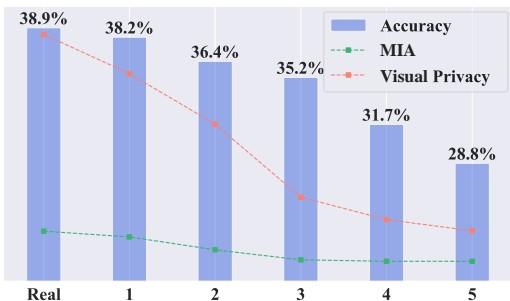

Figure 6: **Impact of KT Parameter $n$ on Privacy and Utility.** Varying $n$ affects visual privacy and data utility.

### 4.4 ROBUSTNESS AGAINST INITIALIZATION-TARGETED ATTACKS

To validate the severity of initialization leakage and the effectiveness of KT protection, we conducted two targeted attacks specifically on the initialization samples using TinyImageNet with `IPC = 50`.

**Nearest-Neighbor Retrieval.** To quantify visual leakage, we executed a nearest-neighbor retrieval attack by performing an LPIPS-based similarity search within the private dataset $\mathcal{T}$ for each distilled image in $\mathcal{S}$. As detailed in Table 3, baseline methods (e.g., DATM with IPC=50) exhibit a critical vulnerability, yielding a Top-1 retrieval success rate of 99.2%. This implies that attackers can reliably recover specific private images used for initialization. In contrast, the integration of KT substantially mitigates this risk, dropping the Top-1 success rate to 18.8% and effectively severing the direct visual linkability between distilled and private samples.

**Fix-Target MIA.** We further assessed membership inference risks specifically targeting the initialization subset, which represents the most vulnerable component of the training data. We conduct experiments on samples both w/o and w/ our proposed KT during initialization, as displayed in Table 4. We choose the maximum value of TPR-FPR as our threshold, and then determine whether a given sample belongs to a member based on this threshold, achieving the MIA success rate.

The results clearly indicate that use original data in DATM significantly leaks membership information of the initial samples. In contrast, KT-DATM *effectively preserves initial private data membership information while simultaneously maintaining generalization.*

Table 3: Nearest-neighbor retrieval attack success rates on Tiny-ImageNet. Lower is better.

| IPC | Method | Top-1 | Top-3 |
|-----|--------|-------|-------|
| 1 | DATM | 18.4% | 36.8% |
| | KT-DATM | 4.2% | 10.6% |
| 10 | DATM | 34.6% | 57.4% |
| | KT-DATM | 10.4% | 24.4% |
| 50 | DATM | 99.2% | 100.0% |
| | KT-DATM | 18.8% | 37.4% |

Table 4: Membership inference on the **initial original samples** in TinyImageNet with `IPC = 50`.

| Method | MIA ASR ($\downarrow$) | Accuracy ($\uparrow$) |
|--------|------------------------|------------------------|
| MTT | 98.8% | 28.0% |
| KT-MTT | 53.4% | 24.7% |
| DATM | 99.5% | 38.6% |
| KT-DATM | 54.1% | 35.2% |
| RDED | 99.4% | 49.6% |
| KT-RDED | 55.6% | 47.3% |

## 5 CONCLUSION

In this study, we first identify that the distilled datasets produced by state-of-the-art distillation methods strongly resemble to original data, indicating significant visual privacy leakage. We identify that the primary source of privacy leakage in distilled data is traced to the initialization of distilled images using original data. Building on these insights, we propose a plug-and-play module, Kaleidoscopic Transformation (KT), which introduces enhanced perturbations to the selected original data during the initialization phase. Extensive experiments verfied that our method KT is able to ensure the denfense against MIA and visual privacy, while preserving the utility of distilled data.

## ETHICS STATEMENT

This work adheres to the ICLR Code of Ethics and complies with the principles of responsible research conduct. All datasets used in our experiments are publicly available and licensed for research purposes. Our work reveals privacy risks for modern dataset distillation, which can visually leak private initialization data. We formalize this risk as visual privacy and propose the KT plug-in as a direct technical safeguard. We therefore argue that an visual privacy evaluation is a critical and necessary addition to existing responsible release practices, for which our work provides both the framework and a solution.

## REPRODUCIBILITY STATEMENT

We have taken extensive steps to ensure the reproducibility of our work. We provide a comprehensive description of our methodology, with all crucial implementation details thoroughly documented within the paper. Detailed descriptions of the datasets, model architecture, optimization settings, and training protocols are included in the Section 4 of the main paper and Appendix E. Together, these resources are intended to allow researchers to fully replicate our results.

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

# A  RELATED WORK

## A.1  DATASET DISTILLATION

Current solutions can be categorized based on their optimization mechanisms (Lei & Tao, 2023): (1) *Meta-Learning Framework*: Distilled data are considered as hyperparameters, which are optimized in a nested loop according to the distilled-data-trained model's risk with respect to (*w.r.t.*) the original data, including DD (Wang et al., 2018), KIP (Nguyen et al., 2021) and FRePo (Zhou et al., 2022). (2) *Gradient Matching*: Aims to match the network gradients computed by the original dataset and the distilled dataset, including DC (Zhao et al., 2020), DSA (Zhao & Bilen, 2021), and IDC (Kim et al., 2022). (3) *Distribution Matching*: Directly matches the distribution of original dataset and distilled data. Methods in this category includ DM (Zhao & Bilen, 2023), CAFE (Wang et al., 2022), SRe$^2$L (Yin et al., 2023). (4) *Trajectory Matching*: Matches the training trajectories of models trained on original and distilled data over multiple steps. This category includes MTT (Cazenavette et al., 2022) and , DATM (Guo et al., 2024). The above methods are based on optimization. Notably, RDED (Sun et al., 2024) introduces an optimization-free paradigm, which directly crop and select realistic patches from the original data, and then stitch the selected patches into the new images as the distilled dataset. It achieves remarkable performance, particularly with large-scale and high-resolution datasets.

## A.2  LiRA

Specifically, the privacy attack LiRA encompasses three stages. Firstly, the adversary randomly samples $N$ datasets from natural distribution to train shadow models. Therefore, for each data sample, there are $N/2$ shadow models trained on it (*the IN models*) and another $N/2$ that are not trained on it (*the OUT models*). Secondly, the adversary estimates the means $\boldsymbol{\mu}_{\text{in}}, \boldsymbol{\mu}_{\text{out}}$, and the variances $\boldsymbol{\sigma}_{\text{in}}^2, \boldsymbol{\sigma}_{\text{out}}^2$ of model confidence for the IN and OUT models, respectively. Finally, to attack, the adversary queries the victim model $f$ with a target example $(\mathbf{x}, y)$ to estimate the likelihood $\Lambda$, defined as:

$$\Lambda := \frac{p(\text{conf}_{\text{obs}} \mid \mathcal{N}(\boldsymbol{\mu}_{\text{in}}, \boldsymbol{\sigma}_{\text{in}}^2))}{p(\text{conf}_{\text{obs}} \mid \mathcal{N}(\boldsymbol{\mu}_{\text{out}}, \boldsymbol{\sigma}_{\text{out}}^2))} \,, \tag{7}$$

where $\text{conf}_{\text{obs}} = \log\left(f(\mathbf{x})_y / 1 - f(\mathbf{x})_y\right)$ is the confidence of target model $f$ on the test example $(\mathbf{x}, y)$. Here, $f(\mathbf{x})_y$ represents the probability assigned by the target model $f$ to the true membership label $y$ when evaluating the attack test example $\mathbf{x}$.

Note that LiRA determines if a data point was part of the training set by comparing a calculated likelihood score $\Lambda$ to a predetermined threshold $\tau$. If $\Lambda > \tau$, the data point is classified as a member of the training set.

# B  PROOF OF PROPOSITION 1

*Proof.* Suppose a full dataset $\mathcal{T}$ and an adjacent dataset $\mathcal{T}'$ which differ in one sample. Let $\mathcal{M}$ be the random sample mechanism that randomly returns a subset of the data without replacement. Let $\mathcal{S}_0, \mathcal{S}_1$ and $\mathcal{S}$ denote the all subsets in $\mathcal{M}(\mathcal{T}), \mathcal{M}(\mathcal{T}')$ and the joint domain of them respectively. For a random subset $S \in \mathcal{S}$, we have

$$\Pr(\mathcal{M}(\mathcal{T}) = S) = \begin{cases} \frac{1}{\binom{|\mathcal{T}|}{|\mathcal{S}|}}, & S \in \mathcal{S}_0, \\ 0, & \text{otherwise.} \end{cases} \tag{8}$$

$$\Pr(\mathcal{M}(\mathcal{T}') = S) = \begin{cases} \frac{1}{\binom{|\mathcal{T}'|}{|\mathcal{S}|}}, & S \in \mathcal{S}_1, \\ 0, & \text{otherwise.} \end{cases} \tag{9}$$

**case 1** $(|\mathcal{T}'| = |\mathcal{T}| + 1)$**:** Due to $\mathcal{T} \subset \mathcal{T}'$, then we have

$$\Pr(\mathcal{M}(\mathcal{T}) \in \mathcal{S}_0) = 1, \tag{10}$$

$$\Pr(\mathcal{M}(\mathcal{T}') \in \mathcal{S}_0) = \frac{\binom{|\mathcal{T}|}{|\mathcal{S}|}}{\binom{|\mathcal{T}'|}{|\mathcal{S}|}} = \frac{\binom{|\mathcal{T}|}{|\mathcal{S}|}}{\binom{|\mathcal{T}|+1}{|\mathcal{S}|}}. \tag{11}$$

We calculate this case based on the definition of differential privacy.

$$
\begin{aligned}
\Pr(\mathcal{M}(\mathcal{T}) \in \mathcal{S}) &= \Pr(\mathcal{M}(\mathcal{T}) \in \mathcal{S}_0) + \Pr(\mathcal{M}(\mathcal{T}) \in \mathcal{S}/\mathcal{S}_0) \\
&= \Pr(\mathcal{M}(\mathcal{T}) \in \mathcal{S}_0) + 0 \\
&= \Pr(\mathcal{M}(\mathcal{T}') \in \mathcal{S}_0) \cdot \frac{\binom{|\mathcal{T}|+1}{|\mathcal{S}|}}{\binom{|\mathcal{T}|}{|\mathcal{S}|}} \\
&= \Pr(\mathcal{M}(\mathcal{T}') \in \mathcal{S}_0) \cdot \frac{|\mathcal{T}|+1}{|\mathcal{T}|-|\mathcal{S}|+1} \\
&\le \Pr(\mathcal{M}(\mathcal{T}') \in \mathcal{S}) \cdot \frac{|\mathcal{T}|+1}{|\mathcal{T}|-|\mathcal{S}|+1}
\end{aligned}
\tag{12}
$$

**case 2** $(|\mathcal{T}'| = |\mathcal{T}| - 1)$ **:** Due to $\mathcal{T}' \subset \mathcal{T}$, then we have

$$
\Pr(\mathcal{M}(\mathcal{T}) \in \mathcal{S}_1) = \frac{\binom{|\mathcal{T}'|}{|\mathcal{S}|}}{\binom{|\mathcal{T}|}{|\mathcal{S}|}} = \frac{\binom{|\mathcal{T}|-1}{|\mathcal{S}|}}{\binom{|\mathcal{T}|}{|\mathcal{S}|}},
\tag{13}
$$

$$
\Pr(\mathcal{M}(\mathcal{T}') \in \mathcal{S}_1) = 1.
\tag{14}
$$

We calculate this case based on the definition of differential privacy.

$$
\begin{aligned}
\Pr(\mathcal{M}(\mathcal{T}) \in \mathcal{S}) &= \Pr(\mathcal{M}(\mathcal{T}) \in \mathcal{S}_1) + \Pr(\mathcal{M}(\mathcal{T}) \in \mathcal{S}/\mathcal{S}_1) \\
&= \Pr(\mathcal{M}(\mathcal{T}) \in \mathcal{S}_1) + \frac{|\mathcal{S}|}{|\mathcal{T}|} \\
&= \Pr(\mathcal{M}(\mathcal{T}') \in \mathcal{S}_1) \cdot \frac{\binom{|\mathcal{T}|-1}{|\mathcal{S}|}}{\binom{|\mathcal{T}|}{|\mathcal{S}|}} + \frac{|\mathcal{S}|}{|\mathcal{T}|} \\
&= \Pr(\mathcal{M}(\mathcal{T}') \in \mathcal{S}_1) \cdot \frac{|\mathcal{T}|-|\mathcal{S}|}{|\mathcal{T}|} + \frac{|\mathcal{S}|}{|\mathcal{T}|} \\
&\le \Pr(\mathcal{M}(\mathcal{T}') \in \mathcal{S}) \cdot \frac{|\mathcal{T}|-|\mathcal{S}|}{|\mathcal{T}|} + \frac{|\mathcal{S}|}{|\mathcal{T}|}
\end{aligned}
\tag{15}
$$

We combine case 1 and case 2, and we have $e^{\epsilon} = \max(\frac{|\mathcal{T}|+1}{|\mathcal{T}|-|\mathcal{S}|+1}, \frac{|\mathcal{T}|-|\mathcal{S}|}{|\mathcal{T}|}) = \frac{|\mathcal{T}|+1}{|\mathcal{T}|-|\mathcal{S}|+1}$, and $\delta = \max(0, \frac{|\mathcal{S}|}{|\mathcal{T}|}) = \frac{|\mathcal{S}|}{|\mathcal{T}|}$. Therefore, randomly sampling $|\mathcal{S}|$ samples from the original dataset (and using them to initialize the distilled dataset) satisfies $(\ln \frac{|\mathcal{T}|+1}{|\mathcal{T}|-|\mathcal{S}|+1}, \frac{|\mathcal{S}|}{|\mathcal{T}|})$-differential privacy. $\qquad \square$

## C   PROOF OF THEOREM 1

*Proof.* The distribution of individual samples in the distilled dataset can be modeled as a normal distribution.

> **Assumption 1 .** *We assume the data of $\mathcal{T}$ and $\mathcal{S}$ are bounded, i.e.,*
>
> $$
> \exists B > 0, \forall \mathbf{x} \in \mathcal{T} \cup \mathcal{S}, \|\mathbf{x}\|_2 \le B.
> \tag{16}
> $$
>
> *For a particular sample $\mathcal{S}_i^*$ in the distilled dataset, to account for the matching stochasticity, we have*
>
> $$
> \mathbf{s}_i^* \sim \mathcal{N}(\mathbf{s}_i + \frac{1}{|\mathcal{T}|} \sum_{j=1}^{|\mathcal{T}|} \mathbf{x}_j - \frac{1}{|\mathcal{S}|} \sum_{j=1}^{|\mathcal{S}|} \mathbf{s}_j, \boldsymbol{\Sigma}_i).
> \tag{17}
> $$

Suppose a full dataset $\mathcal{T}$ and an adjacent dataset $\mathcal{T}'$ which differ in one sample $\mathbf{x}_{\text{differ}}$, such that $\mathbf{x}_{\text{differ}}$ is not used for initialization. The distilled dataset are $\mathcal{S}$ and $\mathcal{S}'$ and $|\mathcal{S}| = |\mathcal{S}'| \ll |\mathcal{T}|$. The distribution of sample $\mathbf{s}_i^*$ within the distilled dataset can be denoted as $p(\mathbf{s}_i^*) = \mathbb{P}(\mathbf{s}_i^*|\mathcal{T})$ and $q(\mathbf{s}_i^*) = \mathbb{P}(\mathbf{s}_i^*|\mathcal{T}')$.

Due to the difference in $\mathbf{x}_{\text{differ}}$, the privacy variations introduced during the matching process can be represented as KL divergence between the two distributions:

$$
\begin{aligned}
D_{KL}(p \parallel q) &= \frac{1}{2} \left( \text{tr}(\boldsymbol{\Sigma}_i^{-1} \boldsymbol{\Sigma}_i) + (\boldsymbol{\mu}_i' - \boldsymbol{\mu}_i)^T \boldsymbol{\Sigma}_i^{-1} (\boldsymbol{\mu}_i' - \boldsymbol{\mu}_i) \right. \\
&\quad \left. -n - \log \frac{\det \boldsymbol{\Sigma}_i}{\det \boldsymbol{\Sigma}_i} \right) \\
&= \frac{1}{2} (\boldsymbol{\mu}_i' - \boldsymbol{\mu}_i)^T \boldsymbol{\Sigma}_i^{-1} (\boldsymbol{\mu}_i' - \boldsymbol{\mu}_i) \\
&\leq \|\boldsymbol{\mu}_i' - \boldsymbol{\mu}_i\|_2 \cdot \lambda_{\max}(\boldsymbol{\Sigma}_i^{-1}).
\end{aligned} \tag{18}
$$

where $n$ is the dimension of $\mathbf{x}$, $\lambda_{\max}$ is the largest eigenvalue of the covariance matrix $\boldsymbol{\Sigma}$ and

$$
\begin{aligned}
\|\boldsymbol{\mu}_i' - \boldsymbol{\mu}_i\|_2 &= \left\| \frac{1}{|\mathcal{T}| - 1} \sum_{j=1}^{|\mathcal{T}|-1} \mathbf{x}_j - \frac{1}{|\mathcal{T}|} \sum_{j=1}^{|\mathcal{T}|} \mathbf{x}_j \right\|_2 \\
&= \frac{1}{|\mathcal{T}|} \left\| \frac{1}{|\mathcal{T}| - 1} \sum_{j=1}^{|\mathcal{T}|-1} \mathbf{x}_j - \mathbf{x}_{\text{differ}} \right\|_2.
\end{aligned} \tag{19}
$$

According to Assumption 1, we have $\|\mathbf{x}\|_2 \leq B$ for all $\mathbf{x} \in \mathcal{T} \cup \mathcal{S}$. Therefore, we have

$$
\left\| \frac{1}{|\mathcal{T}| - 1} \sum_{j=1}^{|\mathcal{T}|-1} \mathbf{x}_j - \mathbf{x}_{\text{differ}} \right\|_2 \leq \left\| \frac{1}{|\mathcal{T}| - 1} \sum_{j=1}^{|\mathcal{T}|-1} \mathbf{x}_j \right\|_2 + \|\mathbf{x}_{\text{differ}}\|_2 \leq 2B. \tag{20}
$$

From previous analysis, it can be concluded that the KL divergence of the distillation results from adjacent datasets is bounded:

$$
D_{KL}(p \parallel q) \leq \frac{2B}{|\mathcal{T}|} \cdot \lambda_{\max}(\boldsymbol{\Sigma}_i^{-1}). \tag{21}
$$

The total KL divergence of the distilled dataset also can be bounded:

$$
D_{KL}(P \parallel Q) \leq \frac{2B |\mathcal{S}|}{|\mathcal{T}|} \cdot \lambda_{\max}(\boldsymbol{\Sigma}^{-1}). \tag{22}
$$

where $P$ and $Q$ are the joint distributions of the adjacent datasets and $\lambda_{\max}(\boldsymbol{\Sigma}^{-1})$ corresponds to the largest eigenvalue of the covariance matrix across all samples in the distilled dataset. $\qquad\square$

## D  PROOF OF THEOREM 2

*Proof.* As demonstrated in the proof of Theorem 1, $\mathcal{T}$ and $\mathcal{T}'$ are adjacent datasets where $\mathcal{T}' = \mathcal{T} \setminus \mathbf{x}_{\text{differ}}$. In section 3.3, we establish the relationship between the KT-initialized distilled data $\mathbf{s}_i'$ and the initialized real data $\mathbf{s}_i$.

$$
\mathbf{s}_i' = \frac{1}{n} \sum_{j=1}^{n} \left( \circ_{k=1}^{m} T_k^{U_{i,j,k} \leq p_k} \right) (\mathbf{s}_i). \tag{23}
$$

where $n$ is the We model the KT as a additive bounded noise $\bar{\boldsymbol{\epsilon}} = \sum_{j=1}^{n} \boldsymbol{\epsilon}_j$, where $\bar{\boldsymbol{\epsilon}} \sim \mathcal{N}(0, \frac{1}{n}\boldsymbol{\Sigma}_\epsilon)$, thus

$$
\mathbf{s}_i' = \mathbf{s}_i + \bar{\boldsymbol{\epsilon}}_i. \tag{24}
$$

where $n$ represents the number of KT candidate transformation images, and $m$ represents the number of types of transformations. We can obtain the KT distilled dataset, optimized for matching as in Theorem 1, whose distribution can be represented as:

$$
\mathbf{s}_i'^* \sim \mathcal{N}\left(\mathbf{s}_i' + \bar{\boldsymbol{\epsilon}}_i + \frac{1}{|\mathcal{T}|} \sum_{j=1}^{|\mathcal{T}|} \mathbf{x}_j - \frac{1}{|\mathcal{S}|} \sum_{j=1}^{|\mathcal{S}|} (\mathbf{s}_j' + \bar{\boldsymbol{\epsilon}}_j), \boldsymbol{\Sigma}_i + \frac{1}{n}\boldsymbol{\Sigma}_\epsilon\right). \tag{25}
$$

Recall the KL divergence upper bound, we have

$$D_{KL}(P_{\text{KT}} \parallel Q_{\text{KT}}) \leq \frac{2B\,|\mathcal{S}|}{|\mathcal{T}|} \cdot \lambda_{\max}((\boldsymbol{\Sigma} + \frac{1}{n}\boldsymbol{\Sigma}_\epsilon)^{-1}). \tag{26}$$

According to the matrix inversion lemma, for positive definite matrices:

$$\lambda_{\max}((\boldsymbol{\Sigma} + \frac{1}{n}\boldsymbol{\Sigma}\epsilon)^{-1}) < \lambda_{\max}(\boldsymbol{\Sigma}^{-1}). \tag{27}$$

Therefore, we have:

$$D_{KL}(P_{\text{KT}} \parallel Q_{\text{KT}}) < D_{KL}(P \parallel Q). \tag{28}$$

After KT initialization, the distillation difference caused by a single sample difference between adjacent datasets is smaller, thereby providing better differential privacy properties. $\qquad\square$

## E    Exprimental Detials

### E.1    Implementation details of KT.

Our method use transformed data via KT instead of real samples for initialization. Notably, it does not involve modifying any distilling datasets process. Experiments were conducted on a single NVIDIA GeForce RTX 4090 GPU (24GB) for IPC=1, and on four RTX 4090 GPUs for the MTT and DATM algorithms at higher ratios (IPC=10 and 50). Thus, our method is a plug-and-play approach that can be easily integrated into existing dataset distillation methods without requiring further modification. We utilize the source code[2] provided by the authors to obtain distilled data distill with IPC $\in \{1, 10, 50\}$.

### E.2    Hyperparameter Settings.

We provide detailed hyperparameter configurations for our distilled dataset evaluation in Figure 6 . For Kaleidoscopic Transformation (KT), we empirically determined that setting $n = 3$ yields the optimal generalization performance, with probability thresholds for each transformation consistent with the DSA (Zhao & Bilen, 2021).

### E.3    A New MIA Framework for Distilled Datasets

Our membership inference attack framework for distilled datasets addresses the limitations of previous approaches by treating the entire original dataset as potential members. Figure 7 illustrates our unified evaluation method using LiRA, which employs common test samples for training shadow models.

This framework ensures a fair comparison across different distillation methods by using identical test samples and shadow models.

Our framework consists of three main steps:

- **Target Model Training:** We train the target model using the distilled dataset, following the same training procedure across all methods. We utilize the original dataset's training samples, designated as members, while the test set comprises non-members.

- **Shadow Model Training:** We train multiple shadow models, ensuring that each sample is treated as a member for half of the shadow models and as a non-member for the other half. To mitigate the potential impact of data augmentation on privacy, we apply DSA with multiple queries during this phase.

- **Attack Evaluation:** We input test cases into both the target and shadow models, computing scores to determine the attack results.

---

[2]DM and DSA: `https://github.com/VICO-UoE/DatasetCondensation`
MTT: `https://github.com/GeorgeCazenavette/mtt-distillation`
DATM: `https://github.com/NUS-HPC-AI-Lab/DATM`
RDED: `https://github.com/LINs-lab/RDED`

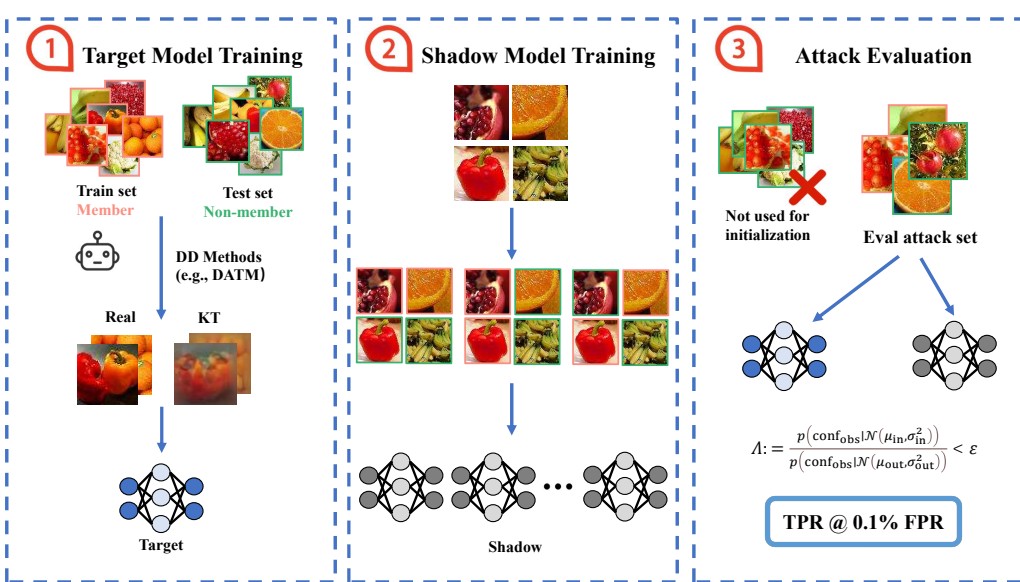

Figure 7: **Unified evaluation method of membership privacy using LiRA:** training shadow models using common test samples.

Table 5: Comparison with previous dataset distillation methods on CIFAR-10. **membership privacy** and **visual privacy** are evaluated via TPR@0.1% FPR and LPIPS, respectively.

| | Method | TPR@0.1%FPR (↓) | | | Min LPIPS (↑) | | | Test Accuracy (↑) | | |
|---|---|---|---|---|---|---|---|---|---|---|
| | | 1 | 10 | 50 | 1 | 10 | 50 | 1 | 10 | 50 |
| CIFAR-10 | Full Dataset | | $8.4 \pm 0.1^*$ | | | $0^*$ | | | $82.24^*$ | |
| | DM | $0.08 \pm 0.02$ | $0.1 \pm 0.02$ | $0.6 \pm 0.05$ | $0.32$ | $0.32$ | $0.15$ | $26.0 \pm 0.8$ | $48.9 \pm 0.6$ | $63.0 \pm 0.4$ |
| | KT-DM | $0.08 \pm 0.02$ | $0.1 \pm 0.03$ | $0.3 \pm 0.03$ | $0.36$ | $0.34$ | $0.32$ | $21.1 \pm 0.3$ | $41.4 \pm 0.4$ | $56.7 \pm 0.4$ |
| | DSA | $0.10 \pm 0.02$ | $0.14 \pm 0.03$ | $1.0 \pm 0.03$ | $0.32$ | $0.24$ | $0.15$ | $26.0 \pm 0.8$ | $48.9 \pm 0.6$ | $63.0 \pm 0.4$ |
| | KT-DSA | $0.10 \pm 0.03$ | $0.12 \pm 0.01$ | $0.18 \pm 0.03$ | $0.33$ | $0.31$ | $0.30$ | $26.0 \pm 0.8$ | $48.9 \pm 0.6$ | $63.0 \pm 0.4$ |
| | MTT | $0.12 \pm 0.01$ | $0.15 \pm 0.01$ | $1.3 \pm 0.1$ | $0.32$ | $0.18$ | $0.09$ | $46.2 \pm 0.8$ | $65.4 \pm 0.7$ | $71.6 \pm 0.2$ |
| | KT-MTT | $0.1 \pm 0.02$ | $0.11 \pm 0.02$ | $0.4 \pm 0.2$ | $0.35$ | $0.33$ | $0.28$ | $42.8 \pm 0.2$ | $59.8 \pm 0.2$ | $66.4 \pm 0.3$ |
| | DATM | $0.13 \pm 0.03$ | $0.26 \pm 0.02$ | $\mathbf{1.6 \pm 0.1}$ | $0.29$ | $0.14$ | $\mathbf{0.01}$ | $46.9 \pm 0.5$ | $66.8 \pm 0.2$ | $76.1 \pm 0.3$ |
| | KT-DATM | $0.1 \pm 0.02$ | $0.14 \pm 0.02$ | $\mathbf{0.4 \pm 0.1}$ | $0.31$ | $0.30$ | $\mathbf{0.26}$ | $43.3 \pm 0.2$ | $62.3 \pm 0.1$ | $69.2 \pm 0.2$ |
| | RDED | $0.14 \pm 0.02$ | $0.27 \pm 0.03$ | $2.0 \pm 0.2$ | $0.02$ | $0.01$ | $0.01$ | $23.3 \pm 0.2$ | $50.2 \pm 0.3$ | $68.4 \pm 0.4$ |
| | KT-RDED | $0.12 \pm 0.01$ | $0.18 \pm 0.03$ | $0.7 \pm 0.1$ | $0.30$ | $0.29$ | $0.26$ | $17.7 \pm 0.2$ | $42.2 \pm 0.2$ | $62.5 \pm 0.3$ |

# F  CIFAR-10 RESULTS IN 4.2

Table 5 presents a comprehensive comparison of our method with previous dataset distillation approaches on the CIFAR-10 dataset. We evaluate performance across three key metrics: membership privacy (measured by TPR@0.1% FPR), visual privacy (measured by Min LPIPS), and dataset utility (measured by Test Accuracy).

# G  COMPARISON OF TRADE-OFFS WITH DP GENERATOR

To comprehensively and fairly compare the privacy protection and data utility tradeoff of KT-DATM with other DP-generators, we conducted more comprehensive experiments on the DP-generators. For the privacy guarantee $\epsilon$, we selected values from $\{1, 5, 10, 20, 50\}$, and obtained the TPR@0.1%FPR and model accuracy under LiRA, as shown in Figure 8. In particular, for PSG, we also conducted experiments with $\epsilon \to \infty$, i.e., without privacy protection by gradient matching noise addition.

It can be observed that as $\epsilon$ is relaxed, the data utility obtained by the DP-generator improves. For PSG, which is a dataset distillation algorithm with DP guarantees, relaxing $\epsilon$ allows it to achieve higher data utility. However, due to its outdated matching paradigm, its performance still lags behind KT-DATM. For DP-MEPF, which only has conditional data generation under DP guarantees, the

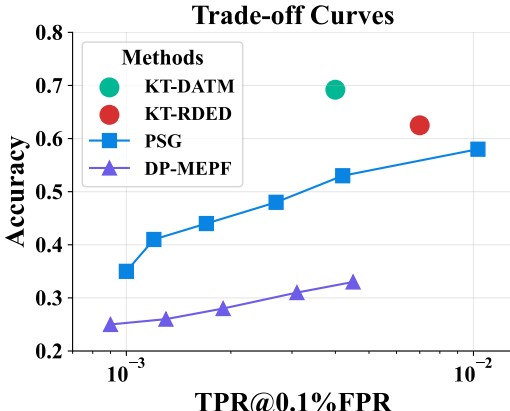

Figure 8: **Trade-off Curves of Privacy Protection and Data Availability for DP-Generators under Different** $\epsilon$. Under consistent protection against MIA, KT-DATM significantly outperforms DP-Generator methods in terms of data utility.

improvement in data utility is limited when relaxing $\epsilon$. However, even when achieving consistent MIA protection, the model accuracy of KT-DATM far exceeds that of PSG and DP-MEPF.

## H  LIMITATIONS

The effectiveness of KT in downstream task accuracy is constrained by the underlying dataset distillation algorithm. While KT can be integrated as a plugin into existing dataset distillation methods to provide efficiency privacy protection, it does not improve the distillation quality from scratch. Our experiments show that KT-RDED outperforms KT-DATM in downstream accuracy, reflecting the base algorithm's capability in preserving task-relevant information. Thus, KT's impact on model performance is inherently tied to the efficacy of the chosen distillation method.

Our method prioritizes mitigating visual leakage over achieving strict $(\epsilon, \delta)$-DP, representing a utility-privacy trade-off. We identify Pure Noise Initialization as a necessary future direction to achieve strict DP without compromising utility.

## I  ADDITIONAL RESULTS ON VISUAL PRIVACY

To demonstrate the robustness of our findings, we have conducted additional experiments using other powerful metrics, SSCD (Pizzi et al., 2022) and DreamSim (Fu et al., 2023), on TinyImageNet with IPC=50. The results are consistent with our original findings. Specifically, we compute a dataset-level visual privacy score as the minimum feature-space distance between any distilled image and any original image (Min SSCD/DreamSim). SSCD and DreamSim features are obtained using their official model checkpoints and default preprocessing. We evaluate the same distilled datasets and report averages over three random seeds. Across methods, KT consistently increases the minimum distances, indicating reduced nearest-neighbor visual alignment, while preserving the relative utility trends among baselines.

## J  ADDITIONAL EXPERIMENTS ON DATASETS AND ARCHITECTURES

### J.1  CROSS-ARCHITECTURE EVALUATION

Our evaluation on CIFAR-10, CIFAR-100, and Tiny-ImageNet follows the standard benchmarks established in the dataset distillation literature. These datasets are widely used by prior work (including MTT, DATM, and DM) as they enable fair comparison and reproducibility.

To validate KT's transferability beyond ConvNet-3/4, we conducted comprehensive cross-architecture experiments on CIFAR-10 (IPC=50). We distilled using DATM on ConvNet backbone and evaluated by training three different architectures: ConvNet, ResNet-18, and VGG-11.

Table 6: Performance of different visual privacy metrics on TinyImageNet with IPC=50.

|  | Min SSCD Distance | Min DreamSim Distance |
|---|---|---|
| DM | 0.14 | 0.1387 |
| KT-DM | 0.29 | 0.2602 |
| MTT | 0.03 | 0.0529 |
| KT-MTT | 0.21 | 0.2474 |
| DATM | 0.04 | 0.0252 |
| KT-DATM | 0.22 | 0.2463 |
| RDED | 0.01 | 0.0049 |
| KT-RDED | 0.19 | 0.2125 |

Table 7: Cross-architecture evaluation on CIFAR-10 (IPC=50). Models are distilled on ConvNet and evaluated on various architectures.

| Test Architecture | DATM (Baseline) | KT-DATM |
|---|---|---|
| ConvNet | 38.6% | 35.2% |
| ResNet-18 | 35.4% | 32.8% |
| VGG-11 | 34.6% | 32.3% |

Results demonstrate that KT maintains consistent visual privacy protection across diverse architectures while preserving competitive utility. The relative accuracy drops remain comparable across architectures, confirming that KT's privacy-utility trade-off generalizes well beyond the distillation backbone.

Due to substantial shadow model training time, cross-architecture MIA results validating membership privacy generalization will be provided during the discussion period.

## J.2 IDENTITY-LEVEL LEAKAGE ON FACE DATA

To directly quantify identity leakage beyond perceptual metrics, we conducted comprehensive experiments on CelebA (IPC=50) using DeepFace (Serengil & Ozpinar, 2024), a state-of-the-art face recognition framework, for identity-aware evaluation.

Table 8: Comparison of perceptual privacy (LPIPS) and utility on CelebA (IPC=50).

| Method | Min LPIPS | Test Accuracy |
|---|---|---|
| Full Dataset | 0 | 95.6% |
| DM | 0.13 | 85.2% |
| KT-DM | 0.27 | 81.8% |
| DATM | 0.01 | 91.3% |
| KT-DATM | 0.23 | 88.7% |

The results in Table 8 demonstrate that baseline distillation methods (DM, DATM) achieve high utility but suffer from extremely low LPIPS scores (e.g., 0.01 for DATM), indicating severe visual similarity to original images. In contrast, applying KT significantly improves visual privacy (increasing LPIPS to 0.23-0.27) with only a marginal drop in accuracy, validating its effectiveness in protecting perceptual privacy for sensitive face data.

**Identity-level privacy evaluation.** We proceed with identity-level privacy evaluation from two perspectives: Face Verification and Face Recognition.

**1. Face Verification (Identity Matching between Distilled and Initialization Samples).** Using DeepFace's verification module, we measured whether distilled images are recognized as the same identity as their initialization samples in Table 9.

Table 9: Identity match rate between distilled images and their initialization samples on CelebA.

| Method | Identity Match Rate |
|--------|---------------------|
| DATM | 100% |
| KT-DATM | 12% |

The baseline DATM exhibits complete identity leakage and distilled image is recognized as the same person as its initialization sample. KT reduces this, demonstrating strong identity-level protection.

**2. Face Recognition (Nearest-Neighbor Retrieval from Original Dataset).** Using DeepFace's feature extractor, we performed nearest-neighbor searches in the original dataset for each distilled image in Table 10 .

Table 10: Nearest-neighbor retrieval success rates using face recognition features on CelebA.

| Method | Top-1 Success Rate | Top-3 Success Rate |
|--------|--------------------|--------------------|
| DATM | 98% | 100% |
| KT-DATM | 8% | 23% |

Baseline DATM allows attackers to nearly perfectly retrieve the original private images through face recognition. KT reduces retrieval success rates, protecting against identity inference attacks.

KT's effectiveness stems from its operation at the initialization phase: it blurs specific facial features and transforms images to appear less face-like, thereby breaking identity linkability. The subsequent trajectory matching process aggregates knowledge from the dataset to recover utility for model training, while the obfuscated identity information remains protected. This synergy between privacy-preserving initialization and knowledge aggregation is key to maintaining the privacy-utility balance.

J.3 ADDITIONAL EXPERIMENTS ON SENSITIVE DATA

Table 11: Performance on COVID-19 Radiography Dataset with IPC = 50.

| | Method | Min LPIPS ($\uparrow$) | Test Accuracy ($\uparrow$) |
|---|--------|------------------------|----------------------------|
| COVID-19 | Full Dataset | 0* | 94.4* |
| | DM | 0.17 | $68.2 \pm 0.4$ |
| | KT-DM | 0.24 | $59.7 \pm 0.3$ |
| | RDED | 0.01 | $87.5 \pm 0.4$ |
| | KT-RDED | 0.24 | $70.7 \pm 0.3$ |

In addition to experiments on benchmark datasets, we conduct experiments on the COVID-19 dataset, specifically for privacy-sensitive scenarios. The COVID-19 Radiography Dataset is a four-class dataset, including COVID, Lung Opacity, Normal, and Viral Pneumonia. We perform experiments with IPC = 50, and the results are presented in Table 11 .

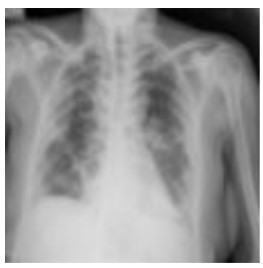 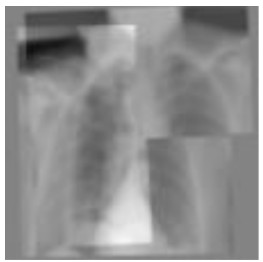

|                 (a) Private                  |                 (b) RDED                  |                 (c) KT-RDED                  |

Figure 9: Samples of the distilled dataset in COVID-19 dataset.

## K    THE USE OF LARGE LANGUAGE MODELS (LLMs)

We employed a large language model, specifically Gemini 2.5 Pro, to assist with proofreading and improving the language and clarity of this paper. All scientific claims, experimental design, results, and conclusions were conceived and articulated by the human authors. The authors have carefully reviewed and edited all text to ensure its scientific accuracy and take full responsibility for the final content of this paper.

