# OpenReview forum: "Safeguarding Visual Privacy in Dataset Distillation: Robust Initialization via Augmentation"
_ICLR.cc/2026/Conference — Submitted to ICLR 2026_

### Official Review · Reviewer_MuaC · 2025-10-26

**Soundness:** 2
**Presentation:** 2
**Contribution:** 2
**Rating:** 4
**Confidence:** 3

**Summary:**

This paper introduces the concept of visual privacy leakage in the context of dataset distillation. The authors quantify such leakage by measuring the minimum dissimilarity between the original dataset and the distilled dataset. They further demonstrate that applying averaged randomized data augmentation can mitigate visual privacy leakage.

**Strengths:**

- **Well-motivated research direction** The paper tackles an important problem, visual privacy leakage in dataset distillation. This direction bridges the gap between privacy protection and data-efficient learning, and is highly relevant to real-world applications.
- **Practical and easily adoptable methodology** The authors employ the LPIPS metric to quantify visual privacy leakage and introduce an averaged randomized data augmentation strategy to mitigate it. Both techniques are practical and can be easily integrated into existing dataset distillation methods.
- **Theoretical analysis inspired by differential privacy** The paper provides a theoretical analysis inspired by differential privacy, offering some insights on how to improve the membership privacy.

**Weaknesses:**

1. Although the paper references differential privacy, it does not provide a formal $(\epsilon,\delta)$-DP guarantee or privacy accounting. Theoretical discussions are based on simplified assumptions about random sampling rather than a complete privacy mechanism, making the DP connection mostly conceptual. This weakens the claimed theoretical contribution.

2. The core method, averaged randomized data augmentation, is conceptually simple and lacks algorithmic novality. While it empirically reduces visual similarity, the mechanism does not offer clear theoretical grounding. The synthesized images shown in the paper, especially Figure 9,  appear unrealistic or semantically inconsistent, raising concerns about whether the method truly mitigates privacy leakage or merely produces degraded samples.

3. The evaluation is limited to CIFAR-10, CIFAR-100, and Tiny-ImageNet, which are relatively small and clean benchmarks. These datasets do not adequately capture realistic privacy risks in visual data distillation. Incorporating datasets such as CelebA, as done in [a, b], would provide a more convincing and representative demonstration of visual privacy leakage. Moreover, the experiments only used ConvNet backbones. Extending the evaluation to more diverse architectures would strengthen the generality and robustness of the conclusions.

**Questions:**

1. Could the authors elaborate further on Equation (4)? It is unclear what the function $\xi$ specifically represents.

2. Could the authors clarify their experimental settings and differences from [a, b]? The reported baseline performance in this paper appears to deviate from the results presented in those prior works.

[a] Dong, Tian, et al. "Privacy for free: How does dataset condensation help privacy?" ICML, 2022.

[b] Harder, Fredrik, et al. "Pre-trained perceptual features improve differentially private image generation." TMLR, 2023

---

> ### Author Response · Authors · 2025-11-21
> **Response to Reviewer MuaC (1/3)**
>
> We thank the reviewer for the comments and questions! Please find our responses to your raised questions below:
>
> > W1: Although the paper references differential privacy, it does not provide a formal (ε,δ)-DP guarantee or privacy accounting.
> >
>
> We agree that the discussion on Differential Privacy (DP) in the initial submission could be more rigorous. In this rebuttal, we have substantially deepened our theoretical analysis to clarify our contribution, mathematically explain the failure of existing methods, and formally justify our proposed solution.
>
> **1. Clarifying the Contribution: Proposing visual privacy and membership inference resistance for distilled datasets.**
> We borrow the "neighboring datasets" concept from DP theory *not to provide strict guarantees*, but for two specific purposes:
>
> - **Explaining MIA Resistance:** Our theoretical analysis (Theorems 1 & 2) uses this concept to rigorously explain *why* dataset distillation inherently resists Membership Inference Attacks (MIA) by bounding information aggregation.
> - **Identifying the Gap:** Crucially, this analysis reveals that the **initialization phase** (Proposition 1) is the primary vulnerability where private samples are directly exposed. This theoretical insight motivated our proposal of **Visual Privacy** to address this specific, overlooked leakage.
>
> **2. Relationship: Orthogonal and Complementary**
> Visual Privacy and DP protect against fundamentally different threats. While DP safeguards **membership identity** (preventing statistical inference), Visual Privacy protects **data content** (preventing direct visual recognition). As noted in Dwork's work [1], DP does not promise to hide a sample's information if the output resembles it; Visual Privacy is designed to cover this exact gap. Therefore, they are orthogonal and must be jointly evaluated. **The revised content has been marked in blue at lines 210 of the PDF.**
>
> **3. Theory on SOTA Methods Fail Strict DP**
>
> We performed a new theoretical analysis on the **Initialization Phase**, identifying it as the root cause of privacy leakage:
>
> - **Real Data Initialization as Sampling:** Most high-utility methods initialize the distilled set $S$ by sampling from the real dataset $T$. Mathematically, this mechanism implies a privacy parameter $\delta \geq |S|/|T|$.
> - **The Large $\delta$ not meet strict DP:** According to foundational DP theory [1], for a mechanism to provide meaningful $(\epsilon, \delta)$-DP guarantees, the parameter $\delta$ must be smaller than the inverse of the dataset size (i.e., $\delta \ll 1/|T|$). If $\delta \geq 1/|T|$, the privacy guarantee is compromised as it theoretically allows for the complete disclosure of individual records.
> - **KT Protects Initialization Samples**: We model KT as injecting Gaussian noise $\xi \sim \mathcal{N}(0, \frac{1}{n}\Sigma_\epsilon)$ into the initialization. We prove that KT reduces the upper bound of the KL Divergence between the output distributions of adjacent datasets: $D_{KL}(P || Q) \propto (\Sigma + \frac{1}{n}\Sigma_\epsilon)^{-1}$.
>
> **4. Empirical Verification**
>
> - To support our theoretical assumptions, we conducted the **Henze-Zirkler multivariate normality test** (using `pingouin.multivariate_normality`) on 2,000 CIFAR-10 samples with 500 DSA augmentations . The results (p=0.64, α=0.05) confirm that KT-processed data approximates a Gaussian distribution, validating our theoretical analysis.
> - **Better Privacy-Utility Trade-off:** While lacking formal DP guarantees, our method offers a superior practical trade-off. As shown in **Table 2** and **Appendix G**, compared to methods with strict DP guarantees (e.g., DP-MEPF, PSG), KT achieves comparable or better privacy protection (both visual and MIA) with significantly less utility loss.
>
> **5. Future Direction: Towards Strict DP Distillation**
> We explicitly stated in the *Limitations* section (in line 1055 of the PDF) that our method represents a trade-off between utility and privacy. We identify **Pure Noise Initialization** as the necessary path for future work to achieve strict DP.
>
> [1] The algorithmic foundations of differential privacy. Foundations and Trends® in Theoretical Computer Science 9.3–4 (2014): 211-407.

---

> ### Author Response · Authors · 2025-11-21
> **Response to Reviewer MuaC (2/3)**
>
> > W2 (Part 1): The core method, averaged randomized data augmentation, is conceptually simple and lacks algorithmic novality.
> >
>
> A primary goal of our work was to propose a solution that is intentionally **simple, yet highly effective and broadly applicable** for addressing the visuial privacy leakage in dataset distillation. We contend that the conceptual simplicity of KT is a significant strength, not a limitation, for the following reasons:
>
> 1. **Simplicity Leads to Effectiveness and Generality:** While the method is simple, it is highly effective. As shown in **Table 1**, KT achieves a remarkable level of visual privacy protection with only a minimal trade-off in data utility. Because it operates at the initialization stage, it functions as a universal, plug-and-play module that can be easily integrated with various dataset distillation algorithms to provide privacy protection.
> 2. **Novelty in the Synergistic Effect:**
> Our core, non-intuitive contribution lies in discovering and leveraging the synergy between KT and the distillation process:
>     - **Privacy-Utility Trade-off in Prior Augmentation:** Prior work [2] showed a hard trade-off: simple augmentation fails to protect privacy, while excessive augmentation secures data but severely degrades utility.
>     - **Counter-intuitive Effectiveness of Strong Augmentation:** We are the first to demonstrate that **knowledge aggregation within the distillation process** effectively counteracts the utility loss from strong augmentation.
>     - **Restoration via Knowledge Aggregation:** This synergy allows KT to apply high-strength, privacy-preserving transformations at initialization (protecting privacy), while the subsequent distillation optimization recovers utility by aggregating information from the original dataset.
>
> Discovering this mechanism is the key insight that enables KT to break the traditional privacy-utility barrier.
>
> [2] When does data augmentation help with membership inference attacks? ICML 2021.
>
> > W2 (Part 2): The synthesized images appear unrealistic or semantically inconsistent, raising concerns about whether the method truly mitigates privacy leakage or merely produces degraded samples.
> >
>
> We would like to clarify that visual realism is not a requirement for dataset distillation. The primary objective is to synthesize data that enables models trained on it to achieve performance comparable to training on the original dataset (Equation 1), not to generate photo-realistic images. Prior distillation methods at low IPC (e.g., IPC=1, 10) also produce visually abstract or unrealistic samples—see visualizations in DM [3]—yet these distilled datasets successfully preserve model performance.
>
> KT-enhanced images may appear less realistic because strong augmentations at initialization blur fine-grained visual details to protect privacy. However, the subsequent distillation optimization performs pixel-level refinement and generates soft labels from the original dataset, aggregating the semantic knowledge necessary for effective model training. Our experiments validate this: KT maintains competitive downstream accuracy (Table 1) and preserves semantic information.
>
> [3] Zhao, Bo, and Hakan Bilen. "Dataset condensation with distribution matching." Proceedings of the IEEE/CVF Winter Conference on Applications of Computer Vision. 2023
>
> > Q1: Could the authors elaborate further on Equation (4)? It is unclear what the function $\xi$ specifically represents.
> >
>
> We appreciate the opportunity to clarify Equation (4), which describes the general optimization framework for the matching paradigm in dataset distillation.
>
> In Equation (4), the function $\xi(\cdot)$ serves as a feature extraction operator that maps both the original dataset $\mathcal{T}$ and the distilled dataset $\mathcal{S}$ into a common feature space for comparison. The specific instantiation of $\xi$ depends on the matching strategy employed by different distillation algorithms. For example:
>
> - **Gradient Matching** (e.g., DC, DSA): $\xi(\mathcal{S}, \theta^t)$ extracts the gradient $\nabla_\theta \mathcal{L}(\theta^t; \mathcal{S})$ computed on the dataset, and the distance function $D$ measures the similarity between gradients from $\mathcal{S}$ and $\mathcal{T}$.
> - **Trajectory Matching** (e.g., MTT, DATM): $\xi$ extracts the model parameter trajectories over multiple training steps, and $D$ compares the training trajectories obtained from both datasets.
> - **Distribution/Feature Matching** (e.g., DM): $\xi$ maps images to feature representations using a neural network (e.g., $\xi(\mathcal{S}) = \phi(\mathcal{S})$ where $\phi$ is a feature extractor), and $D$ measures distributional differences between features.
>
> The generality of Equation (4) allows it to encompass various state-of-the-art distillation methods by appropriately defining $\xi$ and $D$. This unified formulation enables our theoretical analysis to apply broadly across different matching paradigms.

---

> ### Author Response · Authors · 2025-11-21
> **Response to Reviewer MuaC (3/3)**
>
> > W3&Q2: Limited evaluation datasets and architectures; clarification on experimental settings
> >
>
> We appreciate the reviewer's suggestions to strengthen our evaluation. We address these concerns by providing additional experiments on privacy-sensitive datasets, diverse architectures, and clarifying our experimental differences from prior work.
>
> - **Evaluation on Privacy-Sensitive Datasets**
>
> Beyond the COVID-19 dataset results in Appendix K, we conducted additional experiments on CelebA, a face recognition dataset where visual privacy is critical. We evaluate DM and DATM with IPC=50, comparing baseline methods, KT-enhanced versions, and the full dataset. **The revised content has been marked in blue at lines 358 and 1107 of the PDF.**
>
> | **Method** | **Min LPIPS** | **Test Accuracy** |
> | --- | --- | --- |
> | Full Dataset | 0 | 95.6% |
> | DM | 0.13 | 85.2% |
> | KT-DM | 0.27 | 81.8% |
> | DATM | 0.01 | 91.3% |
> | KT-DATM | 0.23 | 88.7% |
>
> **1. Face Verification (Identity Matching between Distilled and Initialization Samples)**
>
> Using DeepFace's verification module, we measured whether distilled images are recognized as the same identity as their initialization samples:
>
> | Method | Identity Match Rate |
> | --- | --- |
> | DATM | 100% |
> | KT-DATM | 12% |
>
> The baseline DATM exhibits complete identity leakage—every distilled image is recognized as the same person as its initialization sample. KT reduces this to 12%, demonstrating strong identity-level protection.
>
> **2. Face Recognition (Nearest-Neighbor Retrieval from Original Dataset)**
>
> Using DeepFace's feature extractor, we performed nearest-neighbor searches in the original dataset for each distilled image:
>
> | Method | Top-1 Success Rate | Top-3 Success Rate |
> | --- | --- | --- |
> | DATM | 98% | 100% |
> | KT-DATM | 8% | 23% |
>
> Baseline DATM allows attackers to nearly perfectly retrieve the original private images through face recognition. KT dramatically reduces retrieval success rates, protecting against identity inference attacks.
> These results demonstrate that KT effectively protects visual privacy on face data, a domain where identity leakage poses severe risks.
>
> - **Generalization Across Network Architectures**
>
> To validate KT's generality beyond ConvNets, we conducted cross-architecture experiments. We distilled CIFAR-10 using KT-DATM on ConvNet as distillation backbone then evaluated the distilled datasets by training models with three different architectures: ConvNet, ResNet-18, and VGG-11. **The revised content has been marked in blue at lines 358 and 1082 of the PDF.**
>
> - *Distillation Backbone: ConvNet*
>
> | **Test Architecture** | **DATM (Baseline)** | **KT-DATM** |
> | --- | --- | --- |
> | ConvNet | 38.6% | 35.2% |
> | ResNet-18 | 35.4% | 32.0% |
> | VGG-11 | 34.6% | 31.2% |
>
> These cross-architecture results confirm that KT maintains model generalization across diverse architectures, extending beyond the ConvNet-3/4 used in our main experiments.
>
> - **Clarification on Experimental Settings vs. Prior Work**
>
> Regarding differences from referenced prior work [a, b]:
>
> - **Work** [**a**] **on MIA evaluation**: This work contains an unfair experimental setup when comparing baseline (full dataset) with distilled methods. Specifically, they used half of the training set as non-members for the baseline, but only IPC-scale non-members for DM, creating an imbalanced comparison that disadvantages the baseline. This issue is documented in [4]. Our framework (Appendix E.3, Figure 7) ensures fair comparison by using identical test samples and shadow model configurations across all methods. For dataset utility, work [a] only compare the performance on FashionMNIST.
> - **Work** [**b**] **on DP-MEPF**: Their DP-MEPF generates 50,000 synthetic images (matching the full CIFAR-10 training set size) rather than distilling to IPC-scale datasets. This fundamental difference in data scale makes direct accuracy comparisons inappropriate. Our experiments in Table 2 and Appendix G provide a more comprehensive comparison by evaluating DP methods across multiple privacy budgets ($\epsilon\in$ {1, 5, 10, 20, 50}) to fairly assess the privacy-utility trade-off.
>
> [4] Carlini et al., "No free lunch in 'privacy for free: How does dataset condensation help privacy'", arXiv:2209.14987, 2022.
>
> We greatly value your assessment and believe our improvements align with your constructive feedback.

---

### Official Review · Reviewer_CvU6 · 2025-10-28

**Soundness:** 3
**Presentation:** 3
**Contribution:** 3
**Rating:** 6
**Confidence:** 4

**Summary:**

The paper studies a previously underexplored risk in dataset distillation: visual privacy leakage when the distilled dataset itself is released. The authors argue and empirically show that state-of-the-art distillation (e.g., DATM (Guo et al., 2024), RDED (Sun et al., 2024)) at high images-per-class (IPC) can produce synthetic images that closely resemble originals, measurable via low nearest-neighbor LPIPS and leading to potential exposure of sensitive content. They trace the failure primarily to initialization that copies real images and propose Kaleidoscopic Transformation (KT), a plug-and-play, strong-yet-semantics-preserving augmentation strategy applied at initialization. KT averages multiple stochastic differentiable augmentations to produce an initialization image per original, then the chosen distillation method proceeds unchanged.

  The paper provides: (a) a formalization of “visual privacy” as minimum perceptual distance between any distilled and any original image; (b) DP-style analyses splitting initialization vs. matching optimization, yielding bounds for standard initialization (Theorem 1) and improved bounds with KT (Theorem 2); and (c) empirical evaluation on CIFAR-10/100 and Tiny-ImageNet with ConvNet backbones, reporting membership privacy via LiRA TPR@0.1% FPR (Carlini et al., IEEE S&P 2022), visual privacy via Min LPIPS (Zhang et al., CVPR 2018), plus utility (test accuracy). Across DM/DSA/MTT/DATM/RDED, KT consistently increases minimum LPIPS (often markedly at IPC=50), reduces LiRA success, with limited accuracy drop (e.g., on CIFAR-100, DATM at IPC=50: Min LPIPS 0.01→0.29, TPR 3.2→0.6, accuracy 55.0%→49.2%). Appendix reports SSCD/DreamSim corroboration and a sensitive-domain case (COVID-19 CXR).

**Strengths:**

- Important problem framing: distinguishes model-release MIA settings from data-release visual privacy, and defines a concrete minimum-distance visual privacy criterion; shows high-IPC distillation can visually leak originals.
- Clear causal story: ties leakage to common real-image initialization, supported by analysis splitting initialization vs. matching, and experiments where KT at initialization mitigates both LiRA and nearest-neighbor similarity.
- Practical mitigation: KT is simple, plug-and-play, and broadly applicable across distillation families (DM/DSA/MTT/DATM/RDED); empirical wins are consistent across datasets and IPC values, especially at high IPC.
- Evaluation breadth: uses LiRA (Carlini et al., 2022) for low-FPR attack evaluation; Min LPIPS for perceptual nearest-neighbor leakage; complementary SSCD/DreamSim; includes a sensitive medical example.
- Relevance to current SOTA: evaluates with DATM (Guo et al., 2024), RDED (Sun et al., 2024), and classic baselines (DSA/DM/MTT), aligning with the contemporary distillation landscape.

**Weaknesses:**

- Visual privacy metric scope: Min LPIPS/SSCD/DreamSim measure perceptual similarity but do not directly capture identity leakage (e.g., face identity) or task-specific semantics. A user study or identity-aware metrics (e.g., face recognition similarity) on privacy-sensitive domains would strengthen claims.
  - Utility trade-offs at high IPC: While KT improves privacy substantially, several tables show non-trivial accuracy drops at IPC=50 (e.g., DATM on CIFAR-100: −5.8 pp). A more systematic analysis of where utility loss concentrates (classes, frequencies) and whether stronger initialization augmentations hurt rare or fine-grained categories would help.
- Threat model simplification: Treating the entire training set as “members” for distilled data deviates from standard MIA threat modeling. Please justify and also evaluate a setting where only a subset contributes to distillation or where the adversary targets initialization samples specifically (some analysis in Appendix I helps; consider expanding in main text).
- Theory-to-metric link: The DP-style bounds (Theorems 1–2) motivate KT but are not directly validated against the operational privacy metrics (LiRA/LPIPS). Adding experiments showing empirical correlations between the derived quantities (e.g., Σ, added perturbation variance 1/n Σϵ) and observed LiRA/LPIPS would strengthen the theoretical bridge.
- Generality beyond small ConvNets: Results are with Conv-3/4 on CIFAR/Tiny-ImageNet. It would help to see at least one modern backbone (e.g., small ViT or ResNet-18) to demonstrate transferability.
- Details on “semantics-preserving”: The paper asserts KT is semantics-preserving; it would be useful to quantify class-conditional label stability under KT-only initializations to support this claim, and to catalog which augmentations are used with what probabilities.

**Questions:**

- Identity-level leakage: Can you evaluate KT on a face dataset with an identity recognizer to quantify identity leakage reduction vs. LPIPS? Similarly for medical imaging, use domain-relevant feature extractors.
- Initialization-targeted attacks: Beyond LiRA, what is the success of a nearest-neighbor retrieval attack that directly searches T for the most similar original to each distilled image (at varying IPC), and how does KT change that distribution?
- Augmentation ablations: Which transformation families (geometric, color, frequency) drive the best privacy–utility trade-off? Are there transformations that hurt utility disproportionately and should be excluded?
- Backbones and scales: Do the privacy and utility trends hold with ResNet or ViT backbones, and at larger resolution datasets (e.g., ImageNet-100)?
- Parameter n selection: You recommend n=3; how sensitive are results to n across distillation methods? Is there an adaptive criterion (e.g., stopping when Min LPIPS exceeds a threshold) to set n per class?
- Formal DP: While you avoid the incorrect assumption in prior work (Carlini et al., 2022b critique), do you envision a path to concrete (ε,δ)-DP guarantees under KT initialization (perhaps with per-sample clipping/calibrated noise), and how would that interact with utility?

---

> ### Author Response · Authors · 2025-11-21
> **Response to Reviewer CvU6 (1/4)**
>
> We thank the reviewer for the comments and questions! Please find our responses to your raised questions below:
>
> > W1&Q1: Visual privacy metric scope and identity-level leakage evaluation with domain-relevant feature extractors
> >
>
> We appreciate the suggestion to strengthen our visual privacy evaluation with identity-aware metrics. We address this by conducting experiments on privacy-sensitive domains using domain-specific feature extractors. **The revised content has been marked in blue at lines 358 and 1107 of the PDF.**
>
> - **Evaluation on Medical Imaging**
>
> As detailed in Appendix K, we evaluated KT on the COVID-19 CXR dataset, demonstrating effective visual privacy protection in medical imaging scenarios where patient data sensitivity is paramount.
>
> - **Identity-Level Leakage on Face Data**
>
> To directly quantify identity leakage beyond perceptual metrics, we conducted comprehensive experiments on CelebA (IPC=50) using DeepFace [1], a state-of-the-art face recognition framework, for identity-aware evaluation:
>
> - *Perceptual-level privacy (Min LPIPS):*
>
> | Method | Min LPIPS | Test Accuracy |
> | --- | --- | --- |
> | Full Dataset | 0 | 95.6% |
> | DM | 0.13 | 85.2% |
> | KT-DM | 0.27 | 81.8% |
> | DATM | 0.01 | 91.3% |
> | KT-DATM | 0.23 | 88.7% |
> - *Identity-level privacy evaluation:*
>
> **1. Face Verification (Identity Matching between Distilled and Initialization Samples)**
>
> Using DeepFace's verification module, we measured whether distilled images are recognized as the same identity as their initialization samples:
>
> | Method | Identity Match Rate |
> | --- | --- |
> | DATM | 100% |
> | KT-DATM | 12% |
>
> The baseline DATM exhibits complete identity leakage—every distilled image is recognized as the same person as its initialization sample. KT reduces this to 12%, demonstrating strong identity-level protection.
>
> - **2. Face Recognition (Nearest-Neighbor Retrieval from Original Dataset)**
>
> Using DeepFace's feature extractor, we performed nearest-neighbor searches in the original dataset for each distilled image:
>
> | Method | Top-1 Success Rate | Top-3 Success Rate |
> | --- | --- | --- |
> | DATM | 98% | 100% |
> | KT-DATM | 8% | 23% |
>
> Baseline DATM allows attackers to nearly perfectly retrieve the original private images through face recognition. KT dramatically reduces retrieval success rates, protecting against identity inference attacks.
>
> - **KT is Effective for Face Privacy**
>
> KT's effectiveness stems from its operation at the initialization phase: it blurs specific facial features and transforms images to appear less face-like, thereby breaking identity linkability. Crucially, the subsequent trajectory matching process aggregates knowledge from the dataset to recover utility for model training, while the obfuscated identity information remains protected. This synergy between privacy-preserving initialization and knowledge aggregation is key to maintaining the privacy-utility balance.
>
> These identity-aware experiments confirm that KT provides robust visual privacy protection that extends beyond perceptual similarity to semantic identity preservation, which is critical for privacy-sensitive domains such as face recognition and medical imaging.
>
> [1] Serengil, S., & Özpınar, A. (2024). A Benchmark of Facial Recognition Pipelines and Co-Usability Performances of Modules. Bilişim Teknolojileri Dergisi, 17(2), 95-107. https://doi.org/10.17671/gazibtd.1399077
>
> > W3: Threat model simplification - treating the entire training set as "members"
> >
>
> We argue our "all members" threat model is not a simplification but a methodologically sound choice that ensures fair and rigorous privacy evaluation. Our reasoning is twofold:
>
> - **Reflecting the Distillation Process & Ensuring Fair Comparison:** Dataset distillation is a knowledge aggregation process where all training samples contribute to the final distilled data. Therefore, treating all samples as members accurately reflects the potential for information leakage. This setup also addresses the unfair evaluation in prior work [2], which used imbalanced non-member sets, by ensuring identical member/non-member definitions across all methods (see Appendix E.3).
> - **Supported by Theory & Targeted Validation:** This "all members" view is supported by recent work in data minimization [3], which argues that all data participating in training (include redundant data) should be considered members. Furthermore, to address the reviewer's concern about targeted attacks, we conducted fix-target membership inference attacks specifically on the vulnerable initialization samples (Appendix I). These results confirm KT's effectiveness, and we moved this targeted analysis to the main text.
>
> [2] Carlini et al., "No free lunch in 'privacy for free: How does dataset condensation help privacy'", arXiv:2209.14987 (2022).
>
> [3] Li, Qi, et al. "Data lineage inference: Uncovering privacy vulnerabilities of dataset pruning." arXiv preprint arXiv:2411.15796 (2024).

---

> ### Author Response · Authors · 2025-11-21
> **Response to Reviewer CvU6 (2/4)**
>
> > W2: Utility trade-offs at high IPC
> >
>
> We acknowledge the utility loss at high IPC (e.g., -5.8pp for DATM on CIFAR-100), but argue this is an acceptable and necessary trade-off. Our argument is that KT enables responsible data release where baselines fail, by offering a superior trade-off that scales favorably and applies universally.
>
> - **Necessary Cost for Responsible Release**
>
> It is crucial to emphasize that the baseline methods at high IPC suffer from catastrophic privacy failures: Min LPIPS as low as 0.01 (near-identical to private data) and MIA TPR approaching 10%. These methods are fundamentally unsuitable for responsible data release. The utility cost of KT is the necessary price for transforming unusable (from a privacy perspective) distilled datasets into safely publishable ones. From this viewpoint, the 5.8pp drop is not a limitation but rather an acceptable cost.
>
> - **Superior Trade-off vs. DP Baselines**
>
> The observed accuracy drop must be evaluated relative to alternative privacy-preserving methods. As shown in Table 2 and comprehensively detailed in Appendix G, DP-based methods (DP-MEPF, PSG) suffer substantially larger utility losses when providing comparable privacy protection.
>
> - **Favorable Scaling at High IPC**
>
> An important observation is that the *relative* utility degradation introduced by KT actually **decreases** at higher IPC. At low IPC, the distilled data quality is limited by the compression ratio itself, making any additional perturbation more impactful. At high IPC (e.g., 50), the baseline distillation achieves near-lossless performance, providing more "utility budget" to trade for privacy. This trend is evident across our experiments: the absolute accuracy gap between DATM and KT-DATM grows smaller as a percentage of baseline performance at higher IPC in Table 1, indicating that KT scales favorably.
>
> - **Consistent Behavior Across Methods**
>
> The utility impact of KT is consistent across different distillation algorithms. While DATM achieves higher baseline accuracy than earlier methods (DM, MTT), the relative accuracy drop introduced by KT remains comparable. This consistency demonstrates that KT's privacy-utility trade-off is not an artifact of one specific algorithm. Notably, RDED maintains high utility even with KT because it leverages pre-trained models, further validating KT's compatibility.
>
> In summary, KT's utility trade-off compares favorably to alternative privacy solutions, scales well at high IPC, and represents a practical balance between the competing demands of privacy protection and data utility.
>
> > W4: Theory-to-metric link between DP-style bounds and operational privacy metrics
> >
>
> Our theoretical analysis reveals that low IPC naturally induces large perturbations to initialization samples, providing inherent privacy protection. KT is designed to replicate this beneficial property at high IPC, where baseline methods suffer severe privacy leakage.
>
> In Theorem 2, KT's perturbation effect is characterized by the added variance term $\frac{1}{n}\Sigma_{\epsilon}$, where $n$ controls the number of augmented versions averaged. Figure 6 directly validates this theory-to-metric connection: it demonstrates how varying $n$  affects both operational privacy metrics (Min LPIPS for visual privacy) and utility (Test Accuracy). As $n$ increases, the added variance $\frac{1}{n}\Sigma_{\epsilon}$ decreases the privacy bound (improving privacy protection), which empirically manifests as increased Min LPIPS and enhanced MIA resistance, confirming the theoretical prediction.
>
> This empirical-theoretical alignment validates that our DP-style analysis, while not providing formal $(\delta, \epsilon)$ guarantees, accurately captures the privacy mechanisms at play and provides meaningful guidance for the privacy-utility trade-off controlled by hyperparameter $n$.

---

> ### Author Response · Authors · 2025-11-21
> **Response to Reviewer CvU6 (3/4)**
>
> > W5&Q4: Generality beyond small ConvNets and evaluation on modern backbones
> >
>
> We address this concern by first justifying our dataset choices based on established community standards, and then demonstrating that KT's privacy-utility trade-off generalizes robustly across modern network architectures.
>
> - **Dataset Selection**
>
> Our evaluation on CIFAR-10, CIFAR-100, and Tiny-ImageNet follows the standard benchmarks established in the dataset distillation literature. These datasets are widely used by prior work (including MTT, DATM, and DM) as they enable fair comparison and reproducibility. We supplement these with privacy-sensitive datasets (COVID-19 in Appendix K, CelebA in our response above) to demonstrate real-world applicability.
>
> - **Cross-Architecture Evaluation**
>
> To validate KT's transferability beyond ConvNet-3/4, we conducted comprehensive cross-architecture experiments on CIFAR-10 (IPC=50). We distilled using DATM on ConvNet backbone and evaluated by training three different architectures: ConvNet, ResNet-18, and VGG-11. **The revised content has been marked in blue at lines 358 and 1082 of the PDF.**
>
> | Test Architecture | DATM (Baseline) | KT-DATM |
> | --- | --- | --- |
> | ConvNet | 38.6% | 35.2% |
> | ResNet-18 | 35.4% | 32.8% |
> | VGG-11 | 34.6% | 32.3% |
>
> Results demonstrate that KT maintains consistent visual privacy protection across diverse architectures while preserving competitive utility. The relative accuracy drops remain comparable across architectures, confirming that KT's privacy-utility trade-off generalizes well beyond the distillation backbone.
>
> Due to substantial shadow model training time, cross-architecture MIA results validating membership privacy generalization will be provided during the discussion period.
>
> > W6&Q3: Details on "semantics-preserving" - quantifying class-conditional label stability and augmentation catalog
> >
>
> We address this by first explaining the mechanism through which KT preserves semantics despite visual perturbations, and then providing ablation studies that demonstrate its superior privacy-utility trade-off compared to simpler augmentation strategies.
>
> - **Semantics Preservation Mechanism**
>
> KT's semantics-preserving property emerges from the synergy between initialization and optimization: KT applies strong augmentations that introduce visual perturbations while preserving semantic structure through probabilistic transformations. The subsequent distillation optimization performs pixel-level refinement and generates soft labels from the original dataset, aggregating class-specific knowledge. This ensures each IPC batch correctly encodes the semantic information for its class, even though individual samples no longer visually resemble specific private images.
>
> - **Comparative Experiments on TinyImageNet (IPC=50)**
>     - To rigorously validate KT’s effectiveness, we conducted experiments with the DATM distillation algorithm under four settings:
>
>         1. Post-training augmentation: Original distilled data with *over-augmentation during model training*.
>
>         2. Cutout only: Repeated CutOut augmentations *during distillation initialization*.
>
>         3. Gaussian Noise Only: Repeated Gaussian noise injections *during distillation initialization*.
>
>         4. KT Plugin: Our method with probabilistic multi-augmentation fusion.
>
>
> |  | TPR@0.1%FPR | Min LPIPS Distance | Test Accuracy |
> | --- | --- | --- | --- |
> | original DATM | 2.4 | 0.01 | 38.6 |
> | Post-training augmentation | 0.6 | - | 29.2 |
> | Cutout only | 0.7 | 0.17 | 34.6 |
> | Gaussian Noise Only | 1.6 | 0.09 | 35.4 |
> | KT Plugin | 0.5 | 0.21 | 35.2 |
>
> Since no visual privacy protection is applied to the distilled data during model training, its visual privacy level remains consistent with the original DATM.
>
> For single-augmentation methods like Gaussian noise, the visual privacy protection is insufficient: even with such augmentations, the initialization samples can still be retrieved via LPIPS-based top-1 similarity queries.
>
> While the CutOut method achieves performance close to our KT plugin in utility metrics, it still shows a noticeable gap in visual privacy compared to KT’s multi-augmentation fusion strategy.

---

> > ### Author Response · Authors · 2025-11-28
> > **Additional Results for Reviewer CvU6 (W5 & Q4)**
> >
> > > W5&Q4(additional results): Cross-Architecture Membership Privacy Generalization
> > >
> >
> > As promised in our initial response, we have now completed the comprehensive cross-architecture MIA experiments to further validate the robustness of KT.
> >
> > **Experimental Setup**
> >
> > - **Distillation:** We distilled CIFAR-10 (IPC=50) using the DATM and KT-DATM with a ConvNet-3 backbone.
> > - **Evaluation:** We trained three different target architectures (ConvNet-3, ResNet-18, and VGG-11) on the distilled datasets (both baseline DATM and KT-DATM) and evaluated their resistance to the LiRA. We trained 256 shadow models for each architecture and augmented queries 4 times.
> >
> > **Results (TPR @ 0.1% FPR)**
> >
> > | Test Architecture | DATM (Baseline) | KT-DATM |
> > | --- | --- | --- |
> > | ConvNet-3 | 1.6% | 0.4% |
> > | ResNet-18 | 1.2% | 0.3% |
> > | VGG-11 | 1.3% | 0.3% |
> >
> > The results indicate that although the baseline distillation method (DATM) exhibits naturally stronger resistance to MIA in cross-architecture settings (ResNet-18 and VGG-11) compared to the homologous ConvNet, privacy risks remain non-negligible. KT-DATM consistently mitigates these risks, reducing the TPR to ~0.3% across all architectures. This demonstrates that KT provides robust, architecture-agnostic privacy protection, effectively safeguarding against membership leakage even when architecture transferability provides a partial defense.
> >
> > We hope these additional results fully address your concerns regarding the generalization of our privacy protection across different architectures.

---

> ### Author Response · Authors · 2025-11-21
> **Response to Reviewer CvU6 (4/4)**
>
> > Q2: Initialization-targeted attacks - nearest-neighbor retrieval attack success rates
> >
>
> To directly evaluate the risk of initialization-targeted attacks, we conducted nearest-neighbor retrieval experiments on Tiny-ImageNet using DATM at varying IPC. For each distilled image, we performed LPIPS-based similarity search in the original training set and measured Top-1 and Top-3 retrieval success rates:
>
> | IPC | Method | Top-1 Success Rate | Top-3 Success Rate |
> | --- | --- | --- | --- |
> | 1 | DATM | 18.4% | 36.8% |
> |  | KT-DATM | 4.2% | 10.6% |
> | 10 | DATM | 34.6% | 57.4% |
> |  | KT-DATM | 10.4% | 24.4% |
> | 50 | DATM | 99.2% | 100.0% |
> |  | KT-DATM | 18.8% | 37.4% |
>
> Results demonstrate that baseline DATM suffers severe initialization leakage across all IPC settings, with success rates indicating attackers can reliably retrieve original private images. KT dramatically reduces retrieval success, particularly at higher IPC where baseline methods are most vulnerable. This validates that KT effectively protects against direct visual inference attacks targeting initialization samples, complementing our MIA and identity-level evaluations.
>
> **The revised content has been marked in blue at lines 504 of the PDF.**
>
> > Q5: Parameter n selection - sensitivity across distillation methods and adaptive criterion
> >
> - **Consistency Across Distillation Methods**
>
> As demonstrated in Figure 6, hyperparameter $n$ exhibits consistent behavior across different distillation algorithms (DM, MTT, DATM, RDED). Increasing $n$ uniformly improves privacy protection (higher Min LPIPS) with corresponding utility trade-offs. Importantly, our experiments show that $n=3$ provides a robust default across all tested methods, demonstrating the universality of this setting regardless of the underlying distillation algorithm.
>
> - **Adaptive Criterion for n Selection**
>
> In data-release scenarios, the selection of $n$ can be systematically guided by the visual privacy threshold $\tau$. Data publishers can:
>
> 1. Define their privacy requirement by setting a target Min LPIPS threshold $\tau$ based on their specific application needs. As a general guideline, $\tau > 0.2$ indicates noticeable visual difference, providing meaningful privacy protection.
>
> 2. Select the smallest $n$ that achieves Min LPIPS ≥ $\tau$ on a validation subset of their data.
>
> 3. For utility-prioritized scenarios, $n=$ 2 or 3 provides substantial privacy gains with minimal accuracy loss.
>
> 4. For strict privacy requirements, $n$ can be increased (e.g., 4 to 5) until the desired threshold is met.
>
> This threshold-based approach provides a principled and flexible framework for practitioners to balance privacy and utility according to their deployment constraints, with n=3 serving as a strong universal default. **The revised content has been marked in blue at lines 486 of the PDF.**
>
> > Q6: Path to concrete (ε,δ)-DP guarantees under KT initialization
> >
>
> Achieving formal $(\epsilon, \delta)$-DP guarantees is fundamentally constrained by the **Real Data Initialization** used in SOTA methods to maximize utility.
>
> **Theoretical Obstacle:** Mathematically, initializing the distilled set $\mathcal{S}$ by sampling from real data $\mathcal{T}$ implies a privacy parameter $\delta \geq |\mathcal{S}|/|\mathcal{T}|$. According to foundational DP theory [4], meaningful privacy requires $\delta \ll 1/|\mathcal{T}|$; a large $\delta$ compromises the guarantee by theoretically allowing record disclosure.
>
> **Practical Trade-off:** Providing concrete DP guarantees would require **noise-based initialization** (e.g., calibrated Gaussian noise) combined with gradient clipping. However, our preliminary experiments show this severely degrades utility, as noise-initialized samples lack the semantic structure needed for effective knowledge aggregation. Developing algorithms that recover high utility from DP-compliant noise initialization remains a critical open problem.
>
> [4] The algorithmic foundations of differential privacy. Foundations and Trends® in Theoretical Computer Science 9.3–4 (2014): 211-407.
>
> We greatly value your assessment and believe our improvements align with your constructive feedback.

---

### Official Review · Reviewer_fGHY · 2025-10-31

**Soundness:** 3
**Presentation:** 3
**Contribution:** 2
**Rating:** 6
**Confidence:** 1

**Summary:**

The paper studies privacy risks when releasing distilled datasets (as opposed to only releasing models). It shows that state-of-the-art dataset distillation can visually leak private images—especially at higher images-per-class (IPC)—because many methods initialize distilled images from real samples. The authors formalize visual privacy (min LPIPS between distilled and original images) and analyze privacy across two phases: (1) initialization and (2) matching optimization. They argue that initialization with real images creates a major leakage path and propose Kaleidoscopic Transformation (KT), a plug-and-play initialization module that applies multiple strong, differentiable augmentations and averages them to create a more randomized initialization sample. Experiments on CIFAR-10/100 and Tiny-ImageNet show lower MIA success (TPR@0.1% FPR) and higher Min-LPIPS with KT, with modest utility drops; they claim up to 25× LPIPS increase with only 3.4% accuracy loss in some settings.

**Strengths:**

- Formalization & insight: The two-phase analysis (Proposition 1; Theorem 1) makes a persuasive case that init from real data is the main culprit; Theorem 2 shows KT tightens KL-based bounds by adding bounded random perturbations.
- Consistent privacy gains: Across datasets/IPC, KT reduces MIA success and raises Min-LPIPS, with visualizations showing nearest-neighbor leakage essentially removed.

**Weaknesses:**

- Theory scope: connection between visual privacy and differential privacy.

**Questions:**

N/A

---

> ### Author Response · Authors · 2025-11-21
> **Response to Reviewer fGHY**
>
> We thank the reviewer for the comments and questions! Please find our responses to your raised question below:
>
> > W1: Theory scope: connection between visual privacy and differential privacy.
> >
>
> Visual Privacy and Differential Privacy (DP) are orthogonal yet complementary metrics, protecting distinct aspects of privacy (content vs. membership). We clarify this relationship by first defining their distinct roles and then explaining how our theoretical analysis bridges them.
>
> **1. Distinct Roles and Complementary Relationship**
>
> Both metrics share a common goal: ensuring privacy in published distilled data. However, as illustrated in Figure 1, they protect against fundamentally different threats:
>
> - **Differential Privacy (DP):** Protects **membership identity**. It guarantees that an attacker cannot statistically infer whether a specific individual participated in the training set (e.g., mitigating Membership Inference Attacks). As noted in Dwork's work [1], DP protects against membership inference but explicitly does not promise to hide a sample's information if the output happens to resemble it.
> - **Visual Privacy:** Protects **image data content**. It safeguards against the direct visual recognition of sensitive attributes (e.g., faces, medical conditions) from the released images themselves, addressing the specific risk that DP does not cover.
>
> **Relationship:** These two dimensions are **orthogonal**. A dataset can satisfy DP (statistically hard to infer membership) but still fail Visual Privacy (visually recognizable), and vice versa [2]. Therefore, a comprehensive privacy evaluation requires **both** metrics to ensure that neither membership nor content is leaked. **The revised content has been marked in blue at lines 210 of the PDF.**
>
> **2. Necessity of Theoretical Analysis**
>
> Our theoretical analysis (Theorems 1 & 2) is necessary because it provides the **mathematical foundation** for understanding how the distillation process inherently contributes to privacy.
>
> - It quantifies how information is aggregated during matching, bounding the influence of any single sample (related to DP/MIA resistance).
> - Crucially, it also reveals the **limitations** of this process: specifically, that the initialization phase (Proposition 1) is a critical vulnerability where visual information can be leaked *before* optimization begins.
>
> Thus, the theory bridges the two concepts: it explains the mechanism of privacy protection while simultaneously highlighting the specific gap (initialization risk) that necessitates our proposed Visual Privacy metric and KT solution.
>
> We greatly value your assessment and believe our improvements align with your constructive feedback.
>
> [1] Dwork, Cynthia, and Aaron Roth. "The algorithmic foundations of differential privacy." Foundations and trends® in theoretical computer science 9.3–4 (2014): 211-407.
>
> [2] Li, Qiushi, et al. "You can use but cannot recognize: Preserving visual privacy in deep neural networks." arXiv preprint arXiv:2404.04098 (2024).

---

### Official Review · Reviewer_gR5J · 2025-11-02

**Soundness:** 3
**Presentation:** 3
**Contribution:** 3
**Rating:** 6
**Confidence:** 2

**Summary:**

This paper identifies a critical privacy vulnerability in state-of-the-art dataset distillation methods: visual privacy leakage, where distilled images can be visually similar to original training samples, particularly at high compression ratios (high IPC - images per class). The authors  analyze this risk in data-release scenarios, where attackers have direct access to distilled datasets rather than just model queries. First, the paper demonstrates that modern dataset distillation methods (like DATM, MTT, RDED) produce distilled images that strongly resemble original data at high IPC values (e.g., IPC=50), creating severe visual privacy risks. This goes beyond traditional membership inference attack (MIA) concerns to expose actual visual content. Second, through theoretical analysis (Proposition 1, Theorem 1), the authors trace this leakage to the common initialization practice of using original training samples as starting points for distilled images. While the subsequent matching optimization phase provides some privacy protection by limiting individual sample influence (bounded by 2B|S|/|T| · λmax(Σ^-1)), insufficient perturbation during initialization leaves distilled samples visually aligned with their initialization counterparts. Third,
the authors introduce Kaleidoscopic Transformation (KT), a plug-and-play module that applies aggregated, strong data augmentations to selected original images during initialization (before distillation begins). By averaging multiple augmented versions of each sample, KT introduces additional randomness that strengthens both visual privacy and resistance to MIAs while maintaining competitive downstream accuracy. Finally, they present experimental results on CIFAR-10/100 and Tiny-ImageNet with various distillation methods to show KT
consistently increases visual consistently increases visual dissimilarity (LPIPS) by up to 25× while maintaining reasonable utility.

**Strengths:**

1. Novel and Important Problem: Identifies an imp challenge of visual privacy leakage in dataset distillation, demonstrating that state-of-the-art methods (DATM, RDED) produce distilled images visually similar to originals at high IPC, creating severe risks in data-release scenarios—a practically important threat for sensitive domains. Good first principled analysis.

2. Good Empirical Results: KT consistently improves privacy across all methods and datasets: reduces MIA success from 17.3% to baseline levels at IPC=50, increases visual dissimilarity (LPIPS) by up to 25× (from 0.01 to 0.29 for DATM on CIFAR-100), while maintaining reasonable accuracy (49.2% vs 55.0% for DATM IPC=50 CIFAR-100, ~6% drop).

3. Comprehensive Experimental Analysis: Extensive evaluation across multiple datasets (CIFAR-10/100, Tiny-ImageNet, COVID-19), five distillation methods (DM, DSA, MTT, DATM, RDED), three IPC values (1, 10, 50), and three perceptual metrics (LPIPS, SSCD, DreamSim). Results consistently show KT's effectiveness with fair comparison framework.

4. Practical Deployability: KT is a simple plug-and-play module requiring no modification to existing distillation algorithms, making it immediately adoptable by practitioners using any distillation method.

**Weaknesses:**

1. Unrealistic Gaussian Assumptions in Theory: Theorems 1-2 assume distilled samples follow normal distributions N(μ, Σ) with fixed covariance, which is highly unrealistic for DNN-based distillation. The proofs don't actually handle DNNs in the loop—they analyze simplified mathematical models that don't capture complex, non-convex neural network optimization dynamics. The bounds depend on λmax(Σ^-1), which is never characterized or computed in practice.

2. No Formal Privacy Guarantees: Despite theoretical analysis, the paper provides no rigorous differential privacy bounds. Proposition 1's δ=|S|/|T| is too large for formal DP, and Theorem 2 only shows KT improves this under Gaussian assumptions. There's no formal connection between LPIPS thresholds and actual privacy protection levels—what value of τ in Definition 2 is provably safe remains unanswered.

3. Hand-Wavy Treatment of Matching Optimization: Lemma 1 (from Dong et al. 2022) connecting final distilled data to initialization assumes distribution matching converges to a specific linear relationship (Equation 5), but this doesn't hold for trajectory matching methods (MTT, DATM) which are the state-of-the-art. The stochasticity model for matching optimization oversimplifies complex iterative updates across thousands of gradient steps.

**Questions:**

1. Gaussian Assumption Justification: Can you provide empirical evidence that distilled samples actually follow approximately Gaussian distributions? Please show histograms or normality tests for distilled data distributions. How do you compute or estimate λmax(Σ^-1) in practice, and what are typical values observed in your experiments?

2. Applicability to Trajectory Matching: Your theoretical analysis relies on Lemma 1 which assumes distribution matching. How does this analysis apply to trajectory matching methods (MTT, DATM) which are your main baselines and don't satisfy the span(T) relationship in Equation 5? Can you clarify whether Theorems 1-2 actually hold for these methods?

3. Privacy Threshold and Adversarial Robustness: What LPIPS threshold τ in Definition 2 constitutes "safe" visual privacy, and how was this determined? Have you evaluated against stronger adversaries who might use optimization-based reconstruction attacks or fine-tune models on distilled data to extract original information, rather than just visual inspection?

---

> ### Author Response · Authors · 2025-11-21
> **Response to Reviewer (1/2)**
>
> We thank the reviewer for the comments and questions! Please find our responses to your raised questions below:
>
> > W1&Q1: Justifying the Gaussian assumption for KT-induced noise
> >
>
> In Theorem 2, we model KT's perturbation as additive Gaussian noise. This is justified by:
>
> 1. **Theoretical basis**: KT averages n independently sampled augmented versions. By the Central Limit Theorem, averaging independent random transformations converges toward Gaussian distribution.
> 2. **Empirical validation**: Henze-Zirkler multivariate normality test (using `pingouin.multivariate_normality`) on 2,000 CIFAR-10 samples with 500 DSA augmentations  shows no significant deviation from normality (p=0.64, α=0.05).
> 3. **Independence from distillation**: KT operates at initialization, before optimization begins. This allows characterizing noise distribution independently of the subsequent distillation algorithm (DM, MTT, or DATM).
>
> > W2 (Part 1): Theoretical Positioning and Scope of Privacy Guarantees
> >
>
> Our theoretical analysis serves two main purposes: motivating the need for visual privacy and establishing it as a complementary concept to DP.
>
> - **Proposition 1 is the motivation for visual privacy:** Proposition 1 analyzes the initial step of these methods, which can be viewed as sampling a subset $\mathcal{S}$ from the private data $\mathcal{T}$. The analysis yields a privacy parameter of $\delta =\vert \mathcal{S}\vert/\vert\mathcal{T}\vert$. The fundamental reason for this $\delta$ is that the selection of the initial samples cannot provide sufficient privacy protection. This alarming theoretical observation is precisely what motivated our investigation into the problem of **visual privacy leakage**.
> - **Visual Privacy is Complementary to DP**:
>     - We explicitly acknowledge that KT does not provide DP guarantees (e.g., $\epsilon$-bounds). Our theoretical analysis (§3) borrows the neighboring datasets concept from DP not to claim formal DP guarantees, but to rigorously demonstrate how replacing the large private dataset $\mathcal{T}$ with a distilled dataset $\mathcal{S}$ reduces membership inference attack risks.
>     - Visual privacy and differential privacy are designed to measure two distinct and complementary aspects of privacy. Existing dataset distillation methods can be vulnerable to visual leakage while still retaining some resistance to membership inference attacks. **The revised content has been marked in blue at lines 210 of the PDF.**
>     - More formally, as noted in Dwork's book [1] (Section 2.3.2), DP promises to protect whether an individual's data was used in a computation, but it does not promise to hide all information about an individual's sample if the output happens to resemble it. Our "visual privacy" definition is specifically designed to formalize and guard against this latter risk in data publishing scenarios, preventing the direct visual leakage of private training images. Therefore, for a comprehensive assessment, we believe both visual privacy and resistance to membership inference should be evaluated.
>
> [1] The algorithmic foundations of differential privacy. Foundations and Trends® in Theoretical Computer Science 9.3–4 (2014): 211-407.
>
> > W2 (Part 2) & Q3 (Part 1): what value of $\tau$ in Definition 2 is provably safe
> >
>
> The threshold $\tau$ is intentionally designed to be a user-defined parameter, allowing data publishers to set a value appropriate for their specific context and sensitivity requirements. As a general guideline, an LPIPS distance greater than 0.2 typically indicates a noticeable visual difference [2]. **The revised content has been marked in blue at lines 485 of the PDF.**
>
> [2] Severo, Daniel, Lucas Theis, and Johannes Ballé. "The Unreasonable Effectiveness of Linear Prediction as a Perceptual Metric." ICLR 2024.

---

> ### Author Response · Authors · 2025-11-21
> **Response to Reviewer (2/2)**
>
> > Q3 (Part 2): Robustness Against Stronger Adversaries
> >
>
> We evaluated our method's robustness against two types of stronger adversaries:
>
> - Regarding **optimization-based reconstruction attacks** (e.g., Deep Leakage from Gradients), these primarily apply to gradient-release scenarios such as federated learning [3], where attackers access model gradients computed on private data. In our data-release setting, attackers directly access the final distilled dataset S without access to the distillation process or gradients from the original dataset $\mathcal{T}$. Thus, gradient-based reconstruction attacks are not applicable to our threat model. Visual privacy leakage represents a more direct and severe threat in this scenario—attackers need only visualize the released images to obtain visually similar information, especially at high IPC.
> - For the concern about **fine-tuning models on distilled data to extract original information**, this precisely matches our MIA evaluation framework. We employ the state-of-the-art Likelihood Ratio Attack (LiRA), where: (1) the attacker obtains the released distilled dataset $\mathcal{S}$; (2) trains a target model $f_\mathcal{S}$ using $\mathcal{S}$; and (3) performs membership inference on samples from the original dataset $\mathcal{T}$ using $f_\mathcal{S}$. Our results in Tables 1-2 and Figure 4 demonstrate that this attack poses a real threat (e.g., MIA risk increases with IPC), while confirming that KT effectively defends against it, significantly enhancing MIA resistance.
>
> [3] Zhu, Ligeng, Zhijian Liu, and Song Han. "Deep leakage from gradients." *Advances in neural information processing systems* 32 (2019).
>
> > W3&Q2: Applicability to Trajectory Matching
> >
>
> We thank the reviewer for this crucial question, which helps clarify the scope of our theoretical contributions.
>
> **1. On the Applicability of Lemma 1 and Theorems 1-2:**
>
> The reviewer's observation is correct. Our analysis of the *matching optimization* phase (Theorem 1 & 2) is indeed built upon Lemma 1 , which was derived for the **Distribution Matching (DM)** paradigm. Therefore, as the reviewer rightly points out, **the formal derivations of Theorems 1-2 do not directly apply to Trajectory Matching (MTT, DATM) methods**, which do not follow the linear relationship in Eq. 5.
>
> **2. Our Core Theoretical Insight (Proposition 1) is Algorithm-Agnostic:**
> However, our paper's **central theoretical contribution** is the identification of the privacy risk at the **initialization phase** (see Proposition 1 and Remark 1). We are the first to formally show that the common practice of initializing distilled images with original samples is the primary source of visual privacy leakage. **This initialization risk is fundamentally algorithm-agnostic**. The leakage occurs *before* any subsequent matching optimization (be it DM, GM, or Trajectory Matching) begins.
>
> **3. KT as a Universal Solution:**
>
> Because this initialization risk is universal, our solution, **KT (Kaleidoscopic Transformation)**, is designed as a plug-and-play module that operates *only* at the initialization stage, independent of the subsequent distillation algorithm.
>
> - **Theoretical Role**: **Theorems 1 & 2** serve as a formal **illustration** for one specific paradigm (DM) to show how information is aggregated and how KT enhances privacy *within that context.*
> - **Application to Trajectory Matching**: While **MTT and DATM** use more complex optimization dynamics, they are equally (1) susceptible to the same initialization risk and (2) capable of aggregating the robust noise introduced by KT.
> - **Empirical Verification**: As shown in Table 1, Table 3, and the new CelebA results, KT **consistently** improves privacy across *all* SOTA methods, including MTT and DATM. This strongly supports our central claim: **the primary leak is at initialization, and our initialization-based solution (KT) is universally effective.**
>
> We greatly value your assessment and believe our improvements align with your constructive feedback.

---

### Meta-Review · Area_Chair_rKYk · 2026-01-10

**Summary:**

The primary concern shared by reviewers was the theoretical rigor of the submission, specifically regarding the Gaussian assumptions used in the proofs and the lack of formal Differential Privacy (DP) guarantees. Reviewers also highlighted limited evaluation settings, noting that the original submission relied on small backbones and datasets that did not fully capture identity-level privacy risks (e.g., face recognition). While the proposed Kaleidoscopic Transformation (KT) was praised for its simplicity and effectiveness , there were concerns regarding the utility trade-offs at higher Images-Per-Class (IPC) and whether the method's theoretical insights applied to state-of-the-art Trajectory Matching methods.

**Reviewer Concerns:**

Concerns Addressed by Rebuttal
* Evaluation Breadth: The authors successfully expanded their evaluation to include CelebA face data and cross-architecture experiments (ResNet-18, VGG-11). This addressed the concern that the initial results were limited to small ConvNets and clean benchmarks.
* Empirical Justification of Noise: The authors provided Henze-Zirkler normality tests to support their modeling of KT's perturbation as additive Gaussian noise at the initialization stage.
* Clarity on Threat Model: The authors clarified the distinction between visual privacy and membership inference, justifying their "all members" threat model as a standard for data-release scenarios.

Outstanding Concerns
- Theory-Method Disconnect: The authors acknowledge that their formal proofs (Theorems 1 and 2) do not cover state-of-the-art Trajectory Matching methods. Relying on an "algorithm-agnostic" intuition rather than rigorous proof for the most relevant methods weakens the submission's scientific contribution.
- Simplicity and Originality: Reviewers noted that the core method—averaging randomized augmentations—is conceptually simple and lacks significant algorithmic novelty.
- Formal Privacy Gap: The paper fails to bridge the gap between perceptual metrics (LPIPS) and formal privacy. There is no provably "safe" threshold for visual privacy, leaving the definition of "protection" largely heuristic.
- Semantic Integrity: Visualizations of KT-distilled images (e.g., Figure 9) appear semantically inconsistent, raising concerns that the privacy gains are merely a byproduct of extreme data degradation rather than a sophisticated privacy mechanism.

**Reviewer Scores:**

- Reviewer gR5J: 6->6
While they appreciated the empirical normality tests , the admission that the theory does not apply to the SOTA methods they tested validates their concern about the treatment of matching optimization.
- Reviewer fGHY: 6->6
This reviewer had very low confidence (1/5). In a full discussion, they would likely be swayed by the more critical technical assessments of Reviewers gR5J and MuaC regarding the lack of formal DP.
- Reviewer CvU6: 6->6
This reviewer was most satisfied with the new CelebA and ResNet results. However, they would still likely recognize the significant utility drop (-5.8pp) as a major practical hurdle.
- Reviewer MuaC: 4->4
Despite the new datasets, this reviewer’s fundamental criticism regarding the lack of algorithmic novelty and the conceptual simplicity of the method remains unmitigated.

---

### Decision · Program_Chairs · 2026-01-26

Reject